# Elevated levels of FMRP-target MAP1B impair human and mouse neuronal development and mouse social behaviors via autophagy pathway

Yu Guo [1,2], Minjie Shen [1,2], Qiping Dong[1], Natasha M. Méndez-Albelo [1,2], Sabrina X. Huang[1,2], Carissa L. Sirois[1,2], Jonathan Le[1,2], Meng Li[1,2], Ezra D. Jarzembowski[1,2], Keegan A. Schoeller [1,2], Michael E. Stockton[1,2], Vanessa L. Horner[3,4], André M. M. Sousa [1,2], Yu Gao [1,2], Birth Defects Research Laboratory*, Jon E. Levine [2,5], Daifeng Wang [1,6], Qiang Chang [1,7,8] & Xinyu Zhao [1,2] ✉

Fragile X messenger ribonucleoprotein 1 protein (FMRP) binds many mRNA targets in the brain. The contribution of these targets to fragile X syndrome (FXS) and related autism spectrum disorder (ASD) remains unclear. Here, we show that FMRP deficiency leads to elevated microtubule-associated protein 1B (MAP1B) in developing human and non-human primate cortical neurons. Targeted *MAP1B* gene activation in healthy human neurons or *MAP1B* gene triplication in ASD patient-derived neurons inhibit morphological and physiological maturation. Activation of *Map1b* in adult male mouse prefrontal cortex excitatory neurons impairs social behaviors. We show that elevated MAP1B sequesters components of autophagy and reduces autophagosome formation. Both MAP1B knockdown and autophagy activation rescue deficits of both ASD and FXS patients' neurons and FMRP-deficient neurons in ex vivo human brain tissue. Our study demonstrates conserved FMRP regulation of MAP1B in primate neurons and establishes a causal link between MAP1B elevation and deficits of FXS and ASD.

Functional deficiency of fragile X messenger ribonucleoprotein 1 (FMR1) protein (FMRP) leads to fragile X syndrome (FXS)[1,2]. FXS is the most common heritable cause of intellectual disability, as well as the largest single genetic contributor to autism spectrum disorder (ASD)[3]. In addition to characteristic executive function impairment, learning deficit, and social anxiety, about 50% of male and 20% of female FXS patients meet the diagnosis of ASD, and FXS contributes to about 3–6% of the ASD population[4,5]. Despite extensive interest in FMRP, how its deficiency impairs neurodevelopment and behavior remains unclear. FMRP is a brain-enriched RNA binding protein that is highly expressed

[1]Waisman Center, University of Wisconsin-Madison, Madison, WI 53705, USA. [2]Department of Neuroscience, School of Medicine and Public Health, University of Wisconsin-Madison, Madison, WI 53705, USA. [3]Department of Pathology and Laboratory Medicine, School of Medicine and Public Health, University of Wisconsin-Madison, Madison, WI 53705, USA. [4]Wisconsin State Laboratory of Hygiene, University of Wisconsin-Madison, Madison, WI 53705, USA. [5]Wisconsin National Primate Research Center, University of Wisconsin-Madison, Madison, WI 53715, USA. [6]Departments of Biostatistics and Medical Informatics, University of Wisconsin-Madison, Madison, WI 53705, USA. [7]Department of Medical Genetics, University of Wisconsin-Madison, Madison, WI 53705, USA. [8]Department of Neurology, School of Medicine and Public Health, University of Wisconsin-Madison, Madison, WI 53705, USA. *A list of authors and their affiliations appears at the end of the paper. ✉e-mail: xinyu.zhao@wisc.edu

in neurons[6]. Genome-wide binding studies have identified a large number of mRNAs bound by FMRP in mouse brains and in human pluripotent stem cells (hPSCs) differentiated neurons[6]. One consistent conclusion drawn from these studies is that FMRP-bound targets are significantly enriched with genes implicated in ASD. However, the functional significance of FMRP regulation of most of these targets remains unclear and very few FMRP targets have been investigated in human neurons.

The mRNA of microtubule-associated protein 1B (MAP1B) has been identified as an FMRP-bound target in animal models[7,8] for over two decades. However, whether FMRP regulation of MAP1B impacts mammalian brain development and FXS remains unknown. To date, whether FMRP deficiency affects MAP1B protein levels is unclear, and no study has assessed the MAP1B levels in FMRP-deficient human tissue or neurons. Microtubule-associated proteins (MAPs) play important roles in neuronal development and functions[9,10]. MAP1B, a large protein (~270 kilodaltons) encoded by a single copy gene, is one of the first MAPs expressed during development and has the highest levels in the central nervous system among MAPs[11]. MAP1B stabilizes microtubules and is a major component of the neuronal cytoskeleton essential for proper dendritic morphogenesis, synaptic maturation, neuronal migration, and axonal guidance[12,13]. The level of MAP1B decreases postnatally but remains high in brain regions with high neural plasticity, including the hippocampus and the prefrontal cortex (PFC)[14]. A loss of function of MAP1B has significant impacts on neuronal development and is absolutely intolerable in both humans[15] and mice[16,17]. On the other hand, elevated MAP1B protein levels have also been implicated in pathogenic conditions, such as mouse models of FXS[8,18,19] and giant axonal neuropathy[20]. MAP1B is located in 5q13.2 and copy number variants (CNVs) in this region, including both duplication and triplication, have been associated with human diseases[21,22]. However, most reported CNVs in 5q13.2 encompass large genomic regions (Supplementary Data 2)[23,24]. Therefore whether MAP1B gene duplication or triplication contributes to human diseases remains unclear. Importantly, no study has connected 5q13.2 CNVs with either FXS or ASD. To date, only a few studies have directly assessed the effect of elevated MAP1B levels in neurons. Additionally, overexpression of MAP1B in cortical progenitors results in impaired radial migration of developing cortical neurons[25]. Overexpression of MAP1B in cultured neurons leads to impaired neurotransmitter trafficking to the membrane[26] and neuronal death[20,27,28]. However, these overexpression studies have used methods to express either full-length or a fragment of MAP1B at very high levels, which do not represent either physiological or pathological conditions. Therefore, it remains unclear whether elevated expression of MAP1B (named "MAP1B-EE" to distinguish from overexpression) in neurons at the levels found in FMRP-deficient brains[19] or disease-associated copy number variants (CNVs)[23,24] contribute to neuronal deficits and disease pathology.

In this study, we provide evidence to show that FMRP indeed controls MAP1B levels and that such regulation is conserved in primates, by using ex vivo human and rhesus macaque brain slice culture and FXS patient-derived neurons. We demonstrate that targeted MAP1B gene activation in healthy human neurons or MAP1B gene triplication (5q13.2trip) in ASD patient-derived neurons impairs morphological and physiological maturation. In addition, our analysis of PsychENCODE data reveals that MAP1B transcript levels are significantly higher in the PFC of ASD compared to control populations. We show that activation of the endogenous Map1b gene in excitatory neurons within the PFC of mice leads to impaired sociability, a behavioral deficit associated with both FXS and ASD[29]. We further demonstrate that MAP1B-EE sequesters microtubule-associated protein 1 light chain 3 (LC3), which prevents LC3 lipidation and subsequent autophagosome formation. Finally, both genetic reduction of MAP1B and treatment with rapamycin to activate autophagy rescue deficits of both 5q13.2trip ASD patient neurons and FMRP-deficient neurons. Our study unveils a causal link between elevated MAP1B levels and neuronal pathology associated with FXS and ASD and provides a mechanism for potential therapeutic treatment.

## Results

### FMRP deficiency leads to elevated MAP1B levels which impairs dendritic development and excitability of human neurons

FMRP binds MAP1B mRNA in a wide range of cell types, including neurons[30–32]. However, only limited studies have assessed MAP1B protein levels in FMRP-deficient cells[8,18,19]. MAP1B protein levels are elevated in the hippocampus and primary cortical neurons derived from Fmr1-KO mice[19] but no study to date has assessed MAP1B protein levels in FMRP-deficient human neurons. We therefore infected ex vivo human mid-fetal cortical slices with lentivirus expressing small hairpin RNA (LV-shFMR1) to knock down FMRP. Among infected cells, about 87% were MAP2-positive and about 92% were NeuN-positive neurons (Supplementary Fig. 1a, b). We observed a significant increase in MAP1B expression levels in LV-shFMR1 infected cortical neurons compared to control LV-shNC infected cortical neurons (Fig. 1a–c, Supplementary Fig. 1c, d). We detected similarly elevated MAP1B levels in LV-shFMR1 infected neurons in rhesus macaque mid-fetal cortical slices (Supplementary Fig. 1e, f), suggesting that FMRP control of MAP1B protein levels is conserved in primates. In addition, we used published human FXS and control pluripotent stem cells (PSCs)[31,33–36] (Supplementary Data 1) to examine the levels of MAP1B. Quantitative polymerase chain reaction (qPCR) analysis showed that MAP1B mRNA levels were higher in two FXS PSC lines (FXS1, FXS2) compared to two control (Ctrl1, Ctrl6) PSC lines (Supplementary Fig. 1g). MAP1B protein levels were also higher in neurons differentiated from FXS iPSCs compared to controls (Supplementary Fig. 1h–k). Therefore, FMRP deficiency in human developing neurons leads to elevated expression of MAP1B (MAP1B-EE).

We therefore investigated the functional impact of MAP1B-EE on human neurons. To mimic MAP1B-EE, we avoided using an overexpression strategy that could confound data interpretation and instead employed a CRISPR/Cas9-mediated targeted transcriptional activation (dCas9-Activator or dCas9A) strategy that allows for inducible activation of endogenous genes. Using our published strategy[33], we created a doxycycline (Dox) inducible dCas9-Activator (idCas9A-H9) human embryonic stem cells (hESC) line (Supplementary Fig. 2a) by targeting the inactive Cas9 (dCas9) fused with 10XGCN4 together with a fusion cassette of GCN4-binding scFv sequence and p65-HSF transcription activator helper complex[37] to the AAVS1 safe harbor locus in the genome of H9 human ESCs[36]. Five idCas9A-H9 ESC clones (#6, #14, #19, #21, and #26) were identified to contain the correct idCas9A-GFP cassette inserted into the AAVS1 safe harbor locus. To specifically activate the MAP1B gene in human neurons, we designed 5 single guide RNAs (sgRNAs) targeting sequences in the proximal promoter (−101 bp to +182 bp relative to transcription start site) of the human MAP1B gene and found that sgRNA candidate #2 could elevate the endogenous MAP1B mRNA levels to 2- to 3-fold compared to control (sgCtrl) in transfected HEK293T cells (Supplementary Fig. 2b). Therefore, we selected sgMAP1B #2 for subsequent experiments in human neurons. We then differentiated all five idCas9A-H9 hESC clones into dorsal forebrain neural progenitor cells (NPCs) using a dual SMAD inhibition method[31] and infected them with lentivirus expressing sgMAP1B (#2) The idCas9A-H9 hESC clone 6 exhibited a ~2-fold increase in endogenous MAP1B mRNA levels (Supplementary Fig. 2c). The integrity of the idCas9A-H9 hESC clone 6 line was confirmed by karyotyping and immunofluorescence for stem cell markers POU5F1 (also known as OCT4), SOX2, and PODXL, as well as the absence of the neuroepithelial marker PAX6 (Supplementary Fig. 2d, e). Off-target CRISPR mutations at the top five predicted sites in the idCas9A-H9 hESC clone 6 were examined by Sanger sequencing, and no off-target mutation was detected (Supplementary Fig. 2f). The idCas9A-H9 hESC clone 6 line was selected for subsequent experiments.

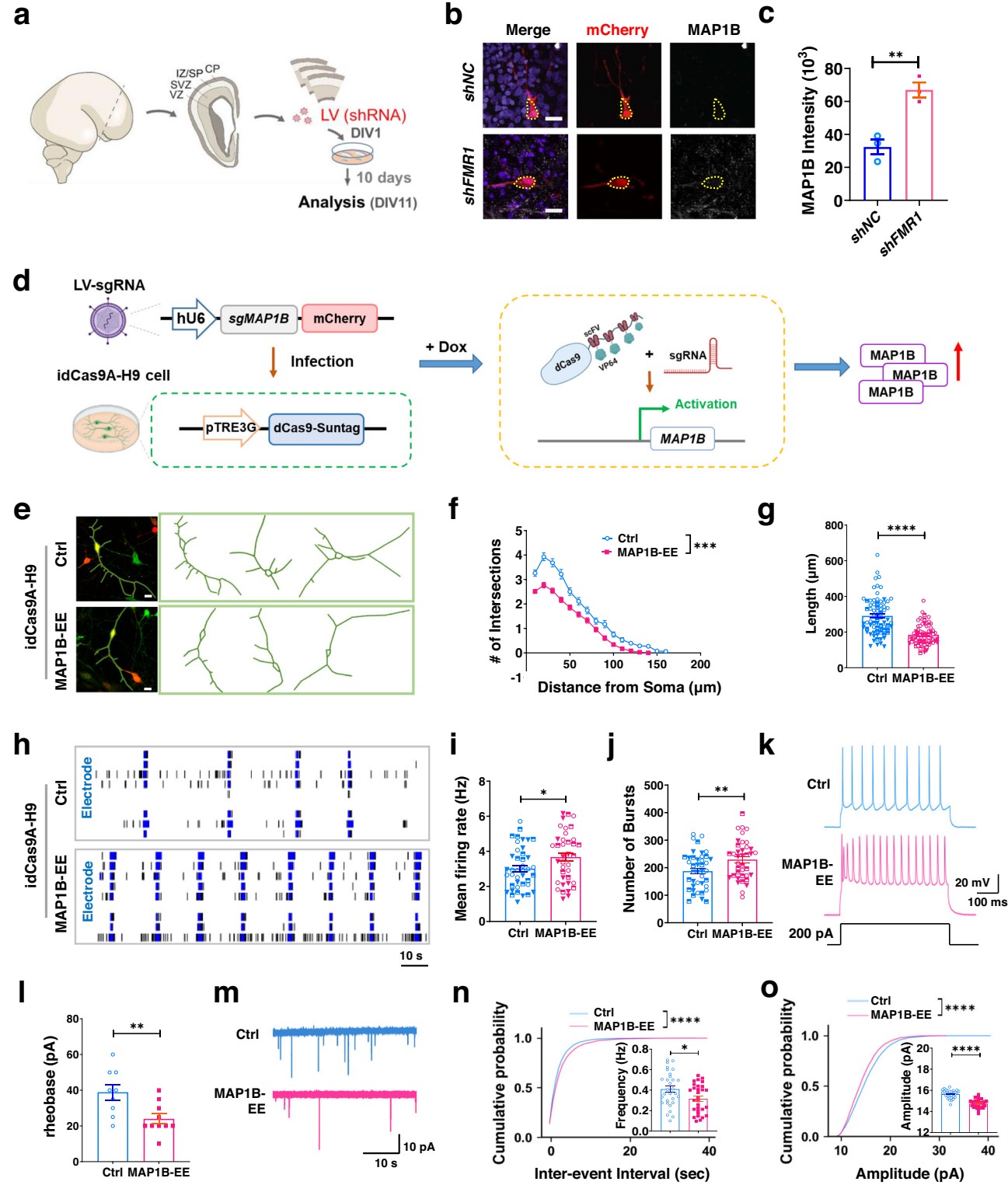

We differentiated idCas9A-H9 NPCs into excitatory neurons using a NGN2-directed differentiation method[38]. MAP1B-EE was achieved by infecting idCas9A-H9 neurons with lentivirus expressing sgRNA (LV-*sgMAP1B*) (Fig. 1d). Both control (LV-*sgCtrl*) and MAP1B-EE (LV-*sgMAP1B*) readily differentiated into highly enriched excitatory neurons confirmed with neuronal markers TUBB3 and VGLUT1 (Supplementary Fig. 2g–i). It has been shown that neurons differentiated from FXS patient-derived iPSCs[34,39] and FXS embryonic stem cells[40,41], as well as neurons isolated from *Fmr1* knockout (KO)

mice[42,43] exhibit reduced dendritic complexity. We therefore assessed the morphology of human neurons with MAP1B-EE. Human neurons with MAP1B-EE also showed significant reductions in both dendritic complexity and total dendritic length compared with controls (LV-*sgCtrl* infected neurons) (Fig. 1e–g). In addition, neurons differentiated from FXS patient-derived iPSCs have been shown to exhibit hyper-excitability measured by multielectrode arrays (MEA) in which extra-cellular electrodes are embedded in cell culture plates to record spontaneous firing[44,45]. We found that human neurons with MAP1B-EE

**Fig. 1 | Elevated MAP1B level leads to impaired dendritic development and altered electrophysiology of human neurons. a** Experimental scheme for assessing MAP1B levels in ex vivo human cortical slices with FMRP knockdown (*shFMR1*). CP cortical plate, SP sub plate, IZ intermediate zone, SVZ subventricular zone, VZ ventricular zone, LV lentivirus, shRNA small hairpin RNA, DIV days in vitro. **b** Representative confocal images of neurons expressing shRNA-mCherry (red), MAP1B (white) in lentivirus-infected cortical slices. Scale bars: 20 μm. **c** MAP1B intensity in mCherry+ neurons in human cortical slices. Two-tailed, unpaired Student's *t*-test, $p = 0.0058$. $N = 3$ individual cortices. **d** Experimental strategy for targeted activation of endogenous *MAP1B* gene expression using an inducible idCas9A-H9 ESC line. DOX Doxycycline. **e** Representative confocal images (from three independent experiments) and traces of GFP+/mCherry+ neurons. Scale bar, 10 μm. **f** Sholl analysis of MAP1B-EE (LV-*sgMAP1B*) and control (LV-*sgCtrl*) neuron. MANOVA, $F_{(1,141)} = 13.990$, $p < 0.001$. Ctrl: $n = 73$ neurons, MAP1B-EE: $n = 70$ neurons from three differentiations, $N = 1$. **g** Total dendritic length. Two-tailed, unpaired Student's *t*-test, $p < 0.0001$. Ctrl: $n = 73$ neurons, MAP1B-EE: $n = 70$ neurons from three differentiations, $N = 1$. **h** Representative raster plots showing 120 s of activity of neurons. Black lines are spikes and blue lines are bursts. **i, j** Quantifications of neuronal mean firing rate, $p = 0.0295$ (**i**) and number of bursts, $p = 0.0087$ (**j**) recorded by MEA. Two-tailed, unpaired Student's *t*-test. Ctrl: $n = 42$ individual wells, MAP1B-EE: $n = 38$ individual wells from three differentiations, $N = 1$. **k** Representative traces showing current injection-evoked action potentials recorded from GFP+/mCherry+ neurons. **l** Quantification of the rheobase current threshold of neurons. Two-tailed, unpaired Student's *t*-test, $p = 0.0087$. Ctrl: $n = 9$ neurons, MAP1B-EE: $n = 10$ neurons from at least five differentiations, $N = 1$. **m** Representative traces of mEPSC recorded from GFP+/mCherry+ neurons. **n, o** Cumulative probability of inter-event intervals and amplitudes of mEPSCs. Two-sided, Mann–Whitney Rank Sum test, $p < 0.0001$. (Inset: mean mEPSC frequency, $p = 0.0221$ and amplitude, $p < 0.0001$. Two-sided, unpaired Student's *t*-test). $n = 30$ neurons from five differentiations, $N = 1$. Error bars: mean ± s.e.m. Source data are provided as a Source Data file.

(LV-*sgMAP1B* infected) also exhibited hyperexcitability compared to control neurons (LV-*sgCtrl* infected), including increased mean firing rate, number of bursts, burst frequency, and network burst duration, without significant changes in burst duration or synchrony index (Fig. 1h–j, Supplementary Fig. 2j–m). To validate MEA data, we performed whole-cell patch-clamp recording to assess the intrinsic excitability of neurons. MAP1B-EE neurons required less injected current to fire an action potential (lower rheobase current) and showed a higher spike frequency in response to current injection compared with the neurons (Ctrl) without elevated MAP1B (Fig. 1k, l), which is consistent with the increased excitability measured by MEA. We further assessed synaptic transmission in these neurons by recording miniature excitatory postsynaptic currents (mEPSCs) and found that MAP1B-EE neurons had decreased frequency (Fig. 1m, n) and amplitude (Fig. 1o) of mEPSCs compared with control neurons, which may result from impaired dendritic maturation. Therefore, MAP1B elevation leads to hyperexcitability and impaired synaptic transmission in human neurons. Taken together, FMRP deficiency leads to MAP1B-EE and MAP1B-EE impairs both morphological and electrophysiological maturation of human neurons.

## MAP1B knockdown rescues neuronal deficits in 5q13.2trip ASD patient iPSC-derived neurons that have MAP1B-EE

*MAP1B* is located in 5q13.2 and CNVs in this region have been associated with human diseases, including neurodevelopmental deficits and autism[21,22] (Supplementary Table 2). However, no study to date has assessed whether duplication or triplication of MAP1B directly contributes to brain disorders. We searched for naturally occurring *MAP1B* gene duplication or triplication cases in human disease databases, including ClinVar[46] and DECIPHER[24], as well as Simons Foundation Autism Research Initiative (SFARI). However, most reported CNVs in 5q13.2 encompass large genomic regions (Supplementary Data 2)[23,24] and a comprehensive assessment of the genes affected by 5q13.2 CNVs for their roles in disease pathology has not been done. Importantly, no study has connected 5q13.2 CNVs with either FXS or ASD. We discovered that an adult male patient diagnosed with ASD at the Waisman Center Clinics has a small 467.5-kilobase (kb) triplication of chromosome 5q13.2 containing the *MAP1B* gene [chr5:71,290,379–71,757,894 based on the Feb 2009 (GRCh37/hg19) assembly], as detected by Illumina microarray analysis of DNA isolated from his skin fibroblasts (Fig. 2a). This triplication was found only in fibroblasts of this 5q13.2trip ASD patient but not his mother (healthy control). No other clinically significant copy number changes or regions of homozygosity were detected in this 5q13.2trip ASD patient. Two additional cases have been reported to have duplication that encompasses the similar 5q13.2 genomic region in patients diagnosed with developmental delay or autism (VCV000153044 and VCV000688929 in ClinVar Database). Three patients with large 5q13.2 duplication or triplication including

the *MAP1B* gene are reported in the DECIPHER Database (Supplementary Data 2). However, cells from these individuals are not available for research. We therefore generated iPSCs from the fibroblasts of the Waisman 5q13.2trip ASD patient.

There are four protein-coding genes in 5q13.2 triplicated region in the Waisman 5q13.2trip ASD patient: *MAP1B*, mitochondria-related *MRPS27* and *PTCD2*, and Zinc finger protein gene *ZNF366* (Fig. 2a). The first three genes were entirely triplicated, but *ZNF366* was partly triplicated (approximately exons 2–5 of 5). qPCR analysis showed that *MAP1B*, *MRPS27*, and *PTCD2* mRNA levels were upregulated but *ZNF366* mRNA level was downregulated in the fibroblasts of the ASD patient, compared to the Control (his mother) (Fig. 2b). Impaired morphogenesis of neuronal dendrites has been associated with both ASD and FXS[39,47]. To determine which of the four genes might contribute to neurodevelopmental deficits, we tested the impact of either up- or downregulation of each gene on the dendritic development of primary neurons isolated from wild-type (WT) mice. We found that neither elevated expression of *Mrps27* or *Ptcd2* nor acute knockdown of *Znf366* through lentivirus (LV)-mediated gene manipulations had significant effect on neuronal dendritic morphology (Supplementary Fig. 3). To activate endogenous *Map1b* gene without excessive overexpression of MAP1B, we again employed a dCas9-Activator strategy, in which dCas9 is fused to transcriptional activation domain VP64 (dCas9-VP64) and co-expressed with a synergistic activation mediator (SAM, MS2-p65-HSF1)[48] (Supplementary Fig. 4a). We designed five sgRNAs targeting the proximal promoter (−305 bp to −24bp relative to transcription start site, TSS) of the mouse *Map1b* gene. Both sgRNA candidates #2 and #5 exhibited strong effects in elevating the endogenous *Map1b* mRNA levels (2- to 3-fold) in transfected Neuro2A cells (Supplementary Fig. 4b). We then transfected WT mouse hippocampal neurons with either *sgMap1b* #2 or #5, together with dCas9-Activator and GFP expression plasmids and analyzed neuronal dendritic morphology. Neurons with *Map1b* gene activation had significant reductions in dendritic complexity and total dendritic length compared with neurons transfected with control sgRNA (*sgCtrl*) (Supplementary Fig. 4c–f). We also observed similar results in primary mouse cortical neurons with *Map1b* gene activation (Supplementary Fig. 4g–i). Therefore, among the four genes affected by 5q13.2trip, MAP1B-EE has the most significant impact on the morphological development of mouse neurons.

We next determined whether reducing the expression levels of MAP1B in 5q13.2trip ASD patient neurons would rescue neuronal deficits. We generated iPSCs from the fibroblasts of the 5q13.2trip ASD patient (Fig. 2c). Integrity of the new ASD iPSC line was confirmed by karyotyping and immunohistological analyses for stem cell markers, OCT3/4, SOX2, and Nanog (Supplementary Fig. 5a, b). ASD iPSCs were differentiated into TUBB3 and VGLUT1-positive cortical excitatory neurons using dual SMAD inhibition method[31] (Fig. 2c, Supplementary Fig. 5c–e). We then investigated the functional impact of genetic

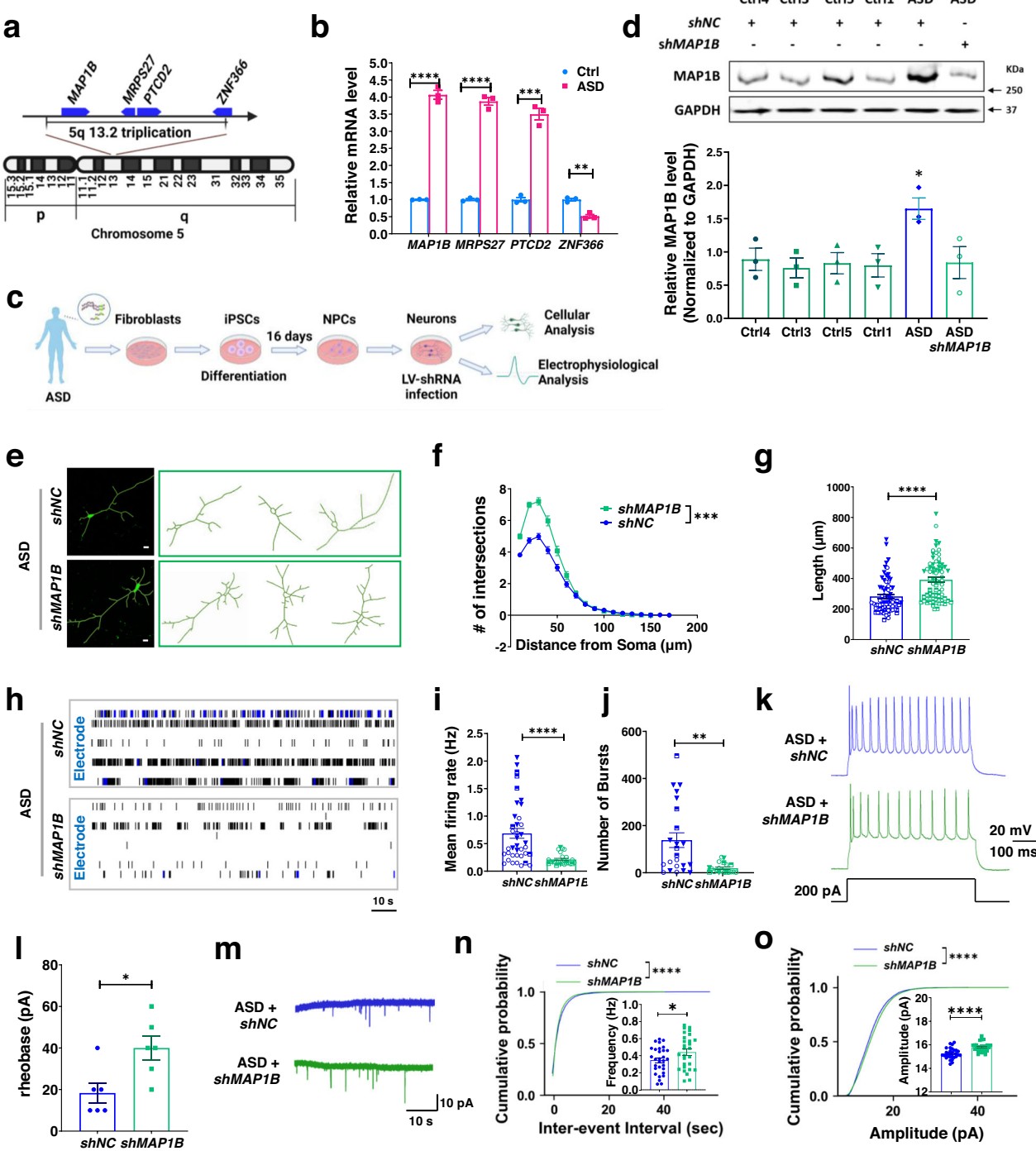

**Fig. 2 | 5q13.2trip ASD patient iPSC-derived neurons exhibit morphological and physiological deficits. a** A schematics showing 5q13.2 triplication in the ASD patient. Blue boxes: genes affected. **b** mRNA levels in fibroblasts of the ASD patient and his mother (Ctrl). Two-tailed, unpaired Student's *t*-test. *MAP1B*: *p* < 0.0001; *MRP27*: *p* < 0.0001; *PTCD2*: *p* = 0.0002; *ZNF366*: *p* = 0.0016. *n* = 3 technical replicates, *N* = 1. **c** Experimental scheme for analyzing ASD patient neurons. **d** Western blot analysis of MAP1B levels. GAPDH: loading control. One-way ANOVA with Dunnett post hoc tests, *p* = 0.0284. **e** Representative confocal images (from three independent experiments) and Neurolucida traces of GFP+ neurons. Scale bar, 10 μm. **f** Sholl analysis of ASD patient neurons with (*shMAP1B*) or without (*shNC*) MAP1B knockdown. MANOVA, *F* (1,140) = 12.001, *p* < 0.001. **g** Total dendritic length. Two-tailed, unpaired Student's *t*-test, *p* < 0.0001. For (**f**) and (**g**), *shNC*: *n* = 70 neurons, *shMAP1B*: *n* = 72 neurons from three differentiations, *N* = 1. **h** Representative raster plots. **i** Neuronal mean firing rate. Two-tailed, unpaired

Student's *t*-test, *p* < 0.0001. Ctrl: *n* = 38 wells, MAP1B-EE: *n* = 25 wells from 3 differentiations, *N* = 1. **j** Number of bursts. Two-tailed, unpaired Student's *t*-test, *p* = 0.0010. Only the wells with bursting activity were analyzed for burst-related parameters, *shNC*: *n* = 23 wells, *shMAP1B*: *n* = 20 wells from three differentiations, *N* = 1. **k** Representative traces showing current injection-evoked action potentials. **l** Rheobase current threshold. Two-tailed, unpaired Student's *t*-test, *p* = 0.0160. *shNC*: *n* = 6 neurons, *shMAP1B*: *n* = 6 neurons from at least four differentiations, *N* = 1. **m** Representative traces of mEPSCs. **n**, **o** Cumulative probability of interevent intervals and amplitudes of mEPSCs. Two-sided, Mann–Whitney Rank Sum test, *p* < 0.0001. (Inset: mean mEPSC frequency, *p* = 0.0472 and amplitude, *p* < 0.0001. Two-sided, unpaired Student's *t*-test). *shNC*: *n* = 30 neurons, *shMAP1B*: *n* = 28 neurons from three differentiations, *N* = 1. Error bars reflect mean ± s.e.m. Source data are provided as a Source Data file.

reduction of MAP1B in ASD patient iPSC-derived neurons using lentivirus expressing shRNA (LV-*shMAP1B*) (Supplementary Fig. 5f, g). Western blot analysis revealed that the MAP1B protein levels were indeed higher in ASD patient neurons compared to those in human neurons differentiated from control human iPSCs (Ctrl1 and Ctrl3) and human ESCs (Ctrl4 and Ctrl5) and LV-*shMAP1B* infection reduced MAP1B levels in ASD neurons to levels similar to those in control neurons (Fig. 2d). ASD neurons with or without MAP1B knockdown readily differentiated into forebrain excitatory neurons (Supplementary Fig. 5c–e). We found that knocking down MAP1B in ASD patient neurons indeed significantly increased the dendritic complexity and total dendritic length compared with controls (LV-*shNC* infected) (Fig. 2e–g).

MEA analysis showed that knocking down MAP1B in ASD neurons also significantly reduced hyper-excitability (Fig. 2h) including mean firing rate, number of bursts, burst frequency, and synchrony index (Fig. 2i, j, Supplementary Fig. 5h, i) but not burst duration (Supplementary Fig. 5j) compared to controls. In addition, by using whole-cell patch-clamp recording, we observed that ASD patient neurons with MAP1B knockdown exhibited an increased rheobase current and a lower frequency of evoked action potentials in response to current injection compared to controls (Fig. 2k, l), which is consistent with the MEA results. Furthermore, the knockdown of MAP1B in patient neurons resulted in increased frequency (Fig. 2m, n) and amplitude (Fig. 2o) of mEPSCs compared with controls. These results demonstrate that MAP1B-EE impairs neuronal dendritic complexity, excitability, and excitatory synaptic transmission in human neurons, which can be corrected by genetic reduction of MAP1B in ASD patient neurons.

## Elevated MAP1B level in excitatory neurons in the PFC leads to ASD-like social abnormities in adult male mice

Although MAP1B is known to be important for dendritic morphogenesis and synaptic maturation[12] and a loss of function of MAP1B has a significant impact on neuronal development in mice[16,17], the impact of MAP1B-EE on behavior and development of diseases is unknown. The discovery of MAP1B triplication in the ASD patient prompted us to investigate whether MAP1B-EE is common among neurodevelopmental and psychiatric disorders. We compared the expression levels of *MAP1B* mRNA transcripts in the PFC (Brodmann Area 9) among the control (CTR), ASD, bipolar disorder (BPD), and schizophrenia (SCZ) populations using transcriptomic data collected by the PsychENCODE consortium[49]. We found that the levels of *MAP1B* transcripts were significantly higher (by ~1.5-fold) in the ASD population, but not in the BPD or the SCZ populations, compared with the CTR population (Fig. 3a), suggesting that MAP1B-EE might be a convergent molecular signature of both FXS and ASD.

We therefore assessed whether increased expression of MAP1B in the PFC might have a direct impact on ASD- and FXS-related behaviors in mice. To achieve targeted activation of endogenous *Map1b* gene in the excitatory neuron of PFC in vivo, we cloned *Map1b* sgRNA #2 (Supplementary Fig. 4b) into an adeno associate viral (AAV) vector (AAV8) to generate recombinant AAV-*sgMap1b* (AAV8-*sgMap1b*-hSyn1-flex-mCherry) virus. We stereotaxically injected AAV-*sgMap1b* (or control AAV-*sgCtrl*) together with AAV-*pCamk2a*-GFP-Cre that expresses Cre recombinase in Calcium/Calmodulin-Dependent Kinase 2 alpha (CAMK2A)-expressing excitatory neurons, into the medial PFC of young adult male Cre-dependent dCas9-Activator mice[37] (Fig. 3b). Mice with targeted activation of the endogenous *Map1b* gene (AAV-*sgMap1b* injected) had 2.2-fold higher MAP1B protein levels in excitatory neurons in the PFC, compared to controls (AAV-*sgCtrl* injected) mice (Fig. 3c, d). We then assessed the behaviors of these mice. There was no significant difference in overall activity levels, anxiety, compulsive behaviors, and spatial learning between experimental groups (Supplementary Fig. 6). However, AAV-*sgMap1b* injected mice showed

significantly reduced interest in another mouse (Social Interest or SI, Fig. 3e, f) and reduced preference of a stranger mouse to familiar one (Social Novelty or SN, Fig. 3e, g) compared to AAV-*sgCtrl* injected mice. Together, these data suggest that MAP1B-EE in PFC excitatory neurons contribute to social behavioral deficits found in FXS and ASD.

## MAP1B elevation leads to autophagy deficiency in neurons

To investigate the molecular mechanism underlying the impacts of MAP1B-EE on neuronal dendritic development and behavior, we first used the BioGRID database to identify proteins with known one-degree physical interactions with MAP1B. We noticed that MAP1B interacts with multiple autophagy-related proteins, including LC3A, LC3B, AMBRA1, ATG3, ATG10, and ATG12, which is supported by literature[50]. Deficits in autophagy have previously been shown in both ASD patients and ASD mouse models[51,52]. We therefore assessed alterations in the autophagy pathway in mouse primary neurons with MAP1B-EE resembling 5q13.2 triplication. Because MAP1B-EE leads to similar morphological deficits in both hippocampal and cortical neurons (Supplementary Fig. 4) and hippocampal neurons consist of mostly pyramidal neurons, therefore are more homogenous than cortical neurons[39], we used mouse hippocampal neurons as the model to study the mechanisms underlying MAP1B-EE. Hippocampal neurons derived from dCas9Activator mice were infected with either LV-*sgMap1b* or LV-*sgCtrl*, together with a lentivirus expressing Cre-GFP (Fig. 4a). We first measured the levels of the autophagosome marker protein LC3[53] and observed lower immunoreactivity for total LC3 levels in MAP1B-EE (*sgMap1b* infected) compared to controls (*sgCtrl* infected) neurons (Fig. 4b, c). The initiation of autophagy depends on the lipidation of cytosolic LC3-I to generate its lipid-modified form LC3-II[54]. Western blot analyses showed that the levels of LC3-II, but not LC3-I, were lower in MAP1B-EE neurons compared to control cells (Fig. 4d, e). Since the mRNA levels of LC3 were not different between MAP1B-EE and controls (Supplementary Fig. 7), decreased LC3-II protein levels could result from either reduced LC3 lipidation affecting autophagosome formation or enhanced LC3-II degradation through autophagosome-lysosome fusion (clearance of autophagosome). Bafilomycin A1 (BafA1) inhibits autophagosome-lysosome fusion, but not autophagosome formation[53,55]. We treated neurons with BafA1 and found that MAP1B-EE led to similarly reduced LC3-II levels with or without BafA1 treatment (Fig. 4f, g), suggesting that MAP1B-EE affects autophagosome formation rather than autophagosome-lysosome fusion. Similar deficits were observed in mouse cortical neurons with MAP1B-EE (Supplementary Fig. 8).

To validate these results, we used a mCherry-GFP-LC3 reporter[56], which expresses pH-insensitive mCherry and pH-sensitive GFP fluorescent proteins, as well as LC3 (Fig. 4h). Hippocampal neurons derived from dCas9Activator mice were transfected with expression plasmids for mCherry/GFP/LC3 reporter and Cre together with expression plasmids for sgRNA (*sgMap1b* or *sgCtrl*) (Fig. 4h). Co-localization of mCherry and GFP signals (yellow) marks autophagosome before fusion with the lysosome. However, upon fusion of autophagosome and lysosome, the GFP signal is quenched by the acidity of the lysosome, and therefore, only mCherry single-positive signal remains (red) (Fig. 4i, j). The mCherry+/GFP+ (yellow) autophagosome puncta and the mCherry-only (mCherry+/GFP−, red) autophagolysosome puncta were captured by a high-resolution confocal microscope, reconstructed in 3D, and quantified. Quantitative analyses showed that MAP1B-EE neurons had reduced numbers of autophagosome (yellow), autophagolysosome (red), and total (red+yellow) puncta compared to controls (Fig. 4k–m), suggesting reduced autophagosome formation in MAP1B-EE. On the other hand, the ratio of autophagosome puncta over total puncta were not significantly different between experimental groups (Fig. 4n), suggesting a normal autophagosome-lysosome fusion in MAP1B-EE. These results support the notion that the autophagy deficits we observed in mouse primary neurons with MAP1B-EE are due to reduced autophagosome formation rather than autophagosome-lysosome fusion. Reduced

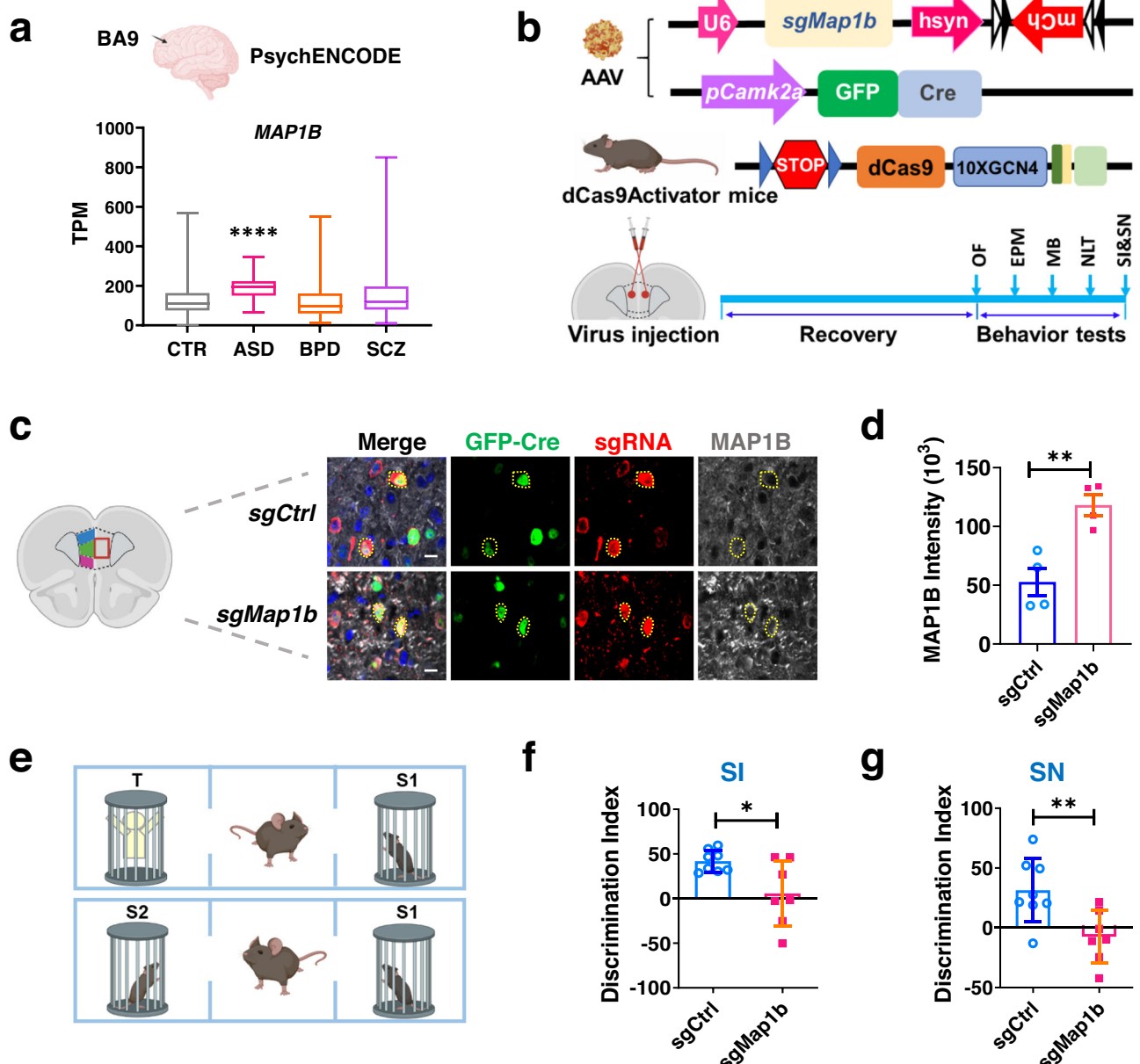

**Fig. 3 | Elevated MAP1B level leads to autism-like behaviors. a** Population gene expression levels of *MAP1B* transcripts in the dorsal lateral prefrontal cortical region (BA9) among several psychiatric disorders and healthy controls collected by PsychENCODE Consortium. *Y*-axis: Gene expression value of *MAP1B* mRNA (RNA-seq TPM). ASD autism spectrum disorder (*n* = 31). BPD bipolar disorder (*N* = 172). SCZ schizophrenia (*N* = 497). CTR, health (*N* = 1166). The gene expression levels of MAP1B in ASD are significantly higher than others by one-side *t*-test. *p* < 8.66e-7 for CTR, *p* < 1.42e-3 for SCZ, and *p* < 4.83e-8 for BPD. **b** Experimental strategy for assessing cognitive functions of targeted *Map1b* gene activation in CAMK2A-expressing excitatory neurons in the PFC of mice. Activation of endogenous *Map1b* gene in the PFC was achieved through stereotaxic injection of AAV expressing *Map1b*-targeting guide RNA (AAV8-*sgMap1b*-hSyn1-flex-mCherry) and AAV expressing Cre driven by *Camk2a* promoter (AAV9-*pCamK2a*-GFP-Cre) into the medial PFC of dCas9Activator mice. **c** Representative confocal images of neurons in the prefrontal cortex (PFC) expressing GFP-Cre (green), sgRNA-mCherry (red), MAP1B (white) in AAV-injected dCas9Activator mice, assessed after behavioral tests. Scale bars: 10 μm. **d** Quantification of MAP1B intensity in GFP+/mCherry+ neurons in mPFC. Two-tailed, unpaired Student's *t*-test, *p* = 0.0045. *N* = 4 individual mice. **e** Experimental scheme of testing social interest (SI) and social novelty (SN). **f, g** MAP1B-EE mice exhibited reduced social interest (SI, **f**), *p* = 0.0195 and social recognition (SN, **g**), *p* = 0.0090. Two-tailed, unpaired Student's *t*-test. *sgCtrl*: *N* = 8 mice, *sgMap1b*: *N* = 7 mice. Data are presented as box plot in (**a**) (center line, median; box limits, upper and lower quartiles; whiskers, min to max). Data are presented as mean ± s.e.m in other panels. Source data are provided as a Source Data file.

autophagosome formation is a common cause of impaired cellular autophagy. Autophagosomes deliver proteins and organelles to lysosomes for degradation by means of cargo adapter proteins, such as p62[51]. When autophagosome formation is reduced, the final degradation step is also reduced, leading to p62 accumulation. We found that the number of p62+ puncta was remarkably increased in MAP1B-EE neurons compared to controls (Fig. 4o, p).

We next assessed the LC3 levels and p62 accumulation in human neurons with MAP1B-EE. In line with the results from mouse neurons (Fig. 4), we observed lower immunoreactivity for LC3 (Fig. 5a, b) and increased p62 accumulation (Supplementary Fig. 9) in human idCas9A-H9 neurons with MAP1B activation (MAP1B-EE) compared to controls. Because FMRP-deficient neurons exhibit MAP1B-EE (Fig. 1, Supplementary Fig. 1), we also assessed the levels of LC3 in neurons

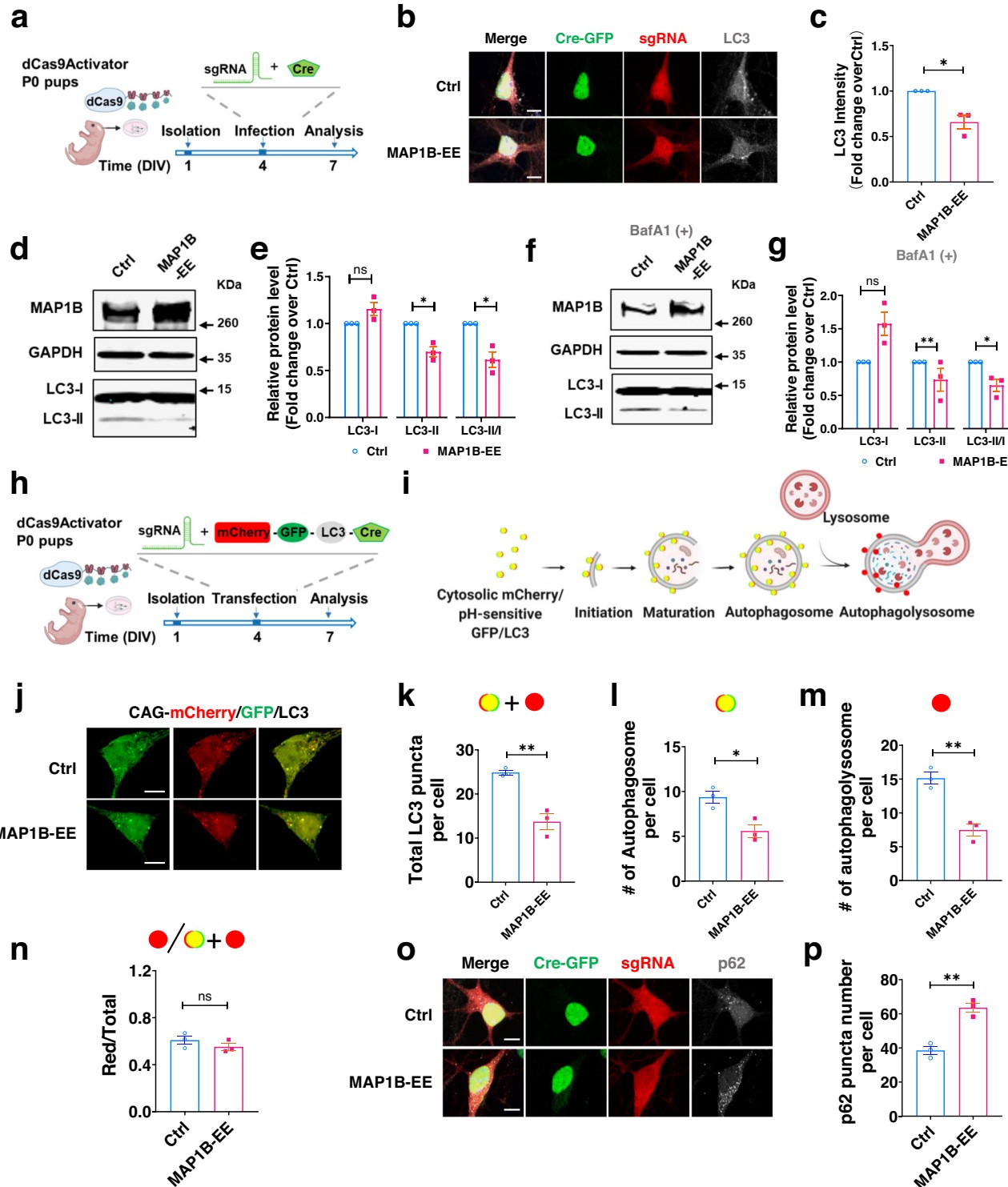

with FMRP knockdown using organotypic cultured brain slices. Noticeably, we found that knocking down FMRP (LV-*shFMR1* infection) led to reduced LC3 levels in both human (Fig. 5c, d) and rhesus macaque (Fig. 5e, f) neurons in mid-fetal cortical slices. In addition, MAP1B knockdown in 5q13.2trip ASD patient neurons resulted in significantly increased LC3 intensity (Fig. 5g, h), higher LC3-II protein levels (Supplementary Fig. 10), reduced p62 accumulation (Supplementary Fig. 11a, b), and increased autophagic activity (Supplementary Fig. 11c, d), compared to controls. Together these results demonstrate that MAP1B-EE observed in pathological conditions, such as FXS and ASD, leads to reduced autophagosome formation in neurons.

## Elevated MAP1B levels sequester cellular LC3 protein and prevent its lipidation in neurons

We next investigated how MAP1B-EE leads to reduced autophagosome formation. The activation of the mammalian target of rapamycin (mTOR) kinase inhibits autophagy by negatively regulating autophagosome formation[57]. We did not detect significant changes in the protein levels of either total mTOR or phosphorylated (activated) mTOR (p-mTOR) in mouse primary neurons with MAP1B-EE (Supplementary Fig. 12), indicating that elevated mTOR activity is not the cause of autophagy deficits resulting from MAP1B-EE. We hypothesized that MAP1B-EE may specifically affect the lipidation of LC3-I to

**Fig. 4 | Elevated MAP1B level leads to autophagy deficits in mouse neurons.**
**a** Experimental scheme for assessing the autophagy in MAP1B-EE neurons. Hippo-
campal neurons were isolated from dCas9Activator mice and infected with LV-Cre-
GFP and LV-sgRNA-mCherry. **b** Representative confocal images of neurons infected
with either *sgMap1b* (MAP1B-EE) or *sgCtrl* (Control) expressing Cre-GFP (green),
sgRNA-mCherry (red), LC3 (white). Scale bars, 10 μm. **c** LC3 intensity. Two-tailed,
unpaired Student's *t*-test with unequal variances, *p* = 0.0476. Ctrl: *N* = 3 indepen-
dent biological replicates. **d** Sample Western blot analysis of mouse neurons with
MAP1B-EE (LV-*sgMap1b* infected) and controls (*sgCtrl*). **e** Quantitative analysis of
LC3-I: *p* = 0.1597; LC3-II: *p* = 0.0297; LC3-II/I (LC3-II/LC3-I): *p* = 0.0434; two-tailed,
unpaired Student's *t*-test with unequal variances. **f** Sample Western blot analysis of
hippocampal neurons with MAP1B-EE and controls treated with BafA1 or vehicle.
**g** Quantitative analysis of proteins treated with BafA1 before harvest. LC3I:
*p* = 0.1359; LC3-II: *p* = 0.0049; LC3-II/I: *p* = 0.0107; two-tailed, unpaired Student's
*t*-test with unequal variances. For (**e**) and (**g**), protein amounts were normalized to
GAPDH and subsequently normalized to control cells. *N* = 3 biological replicates.
**h, i** Experimental scheme for assessing autophagy flux. Hippocampal neurons from
dCas9Activator mice were transfected with an autophagy reporter (CAG-mCherry/
GFP/LC3-IRES-Cre) together with either *sgCtrl* or *sgMap1b*. **j** Representative con-
focal images of autophagy reporter-transfected cells showing autophagosomes
(yellow puncta) and autophagolysosomes (red puncta) in neurons. Scale bars,
10 μm. **k-n** Number of total puncta, *p* = 0.0038 (**k**), autophagosomes, *p* = 0.0188
(**l**), and autophagolysosomes, *p* = 0.0035 (**m**), and calculated ratio of autophago-
lysosomes over total puncta, *p* = 0.3070 (**n**). Quantification of puncta was done
after 3D reconstruction, two-tailed, unpaired Student's *t*-test was used. *N* = 3 bio-
logical replicates. **o** Representative confocal images of p62 (white) puncta localized
in hippocampal neurons. Scale bars, 10 μm. **p** Quantification of p62 puncta was
done after 3D reconstruction. Two-tailed, unpaired Student's *t*-test, *p* = 0.0026.
*N* = 3 biological replicates. All data are presented as mean ± s.e.m. Source data are
provided as a Source Data file.

generate LC3-II. The LC3 lipidation process involves several
autophagy-related proteins (ATG)[58]. During autophagy induction,
ATG4 cleaves the C-terminal peptide in LC3 to form LC3-I, a suitable
substrate for conjugation to phospholipids. Lipidation of LC3-I to form
LC3-II is mediated by ATG7, ATG3, and the ATG12–ATG5–ATG16L1
complex. Hence, we assessed the levels of several proteins involved in
LC3 lipidation in mouse primary neurons and found that MAP1B-EE had
either no effect or a very mild effect on the levels of these proteins
(Supplementary Fig. 13). Therefore, the reduction of LC3-II levels in
MAP1B-EE neurons was unlikely to be a result of altered expression of
proteins involved in LC3 lipidation.

It has been shown that MAP1B can bind LC3 with high affinity[59]. We
hypothesized that elevated MAP1B sequesters LC3-I and prevents its
lipidation, leading to reduced autophagosome formation (Fig. 6a). To
test this model, we determined whether MAP1B-EE could lead to
increased MAP1B-LC3 binding. We expressed equal amounts of GFP-
LC3 fusion protein as bait for MAP1B and ATGs in MAP1B-EE (*sgMAP1B*
transfected) and control (*sgCtrl* transfected) HEK293T cells. We then
pulled down the GFP-LC3 fusion protein and its cellular interacting
proteins using co-immunoprecipitation followed by Western blot
analysis. We found that more MAP1B and less ATG7 were pulled down
with GFP-LC3 fusion protein from MAP1B-EE cells compared to con-
trols (Fig. 6b–d), suggesting increased MAP1B-LC3 interaction but
reduced ATG7-LC3 interaction in MAP1B-EE cells. These results suggest
that MAP1B-EE sequesters cellular LC3-I which reduces LC3 interaction
with lipidation machinery.

To validate these results, we used Proximity Ligation Assays
(PLAs) that allow for detecting protein-protein interaction in proximity
in situ[60]. After 3D reconstruction, we quantified the PLA puncta of
MAP1B-LC3 interaction and ATG7-LC3 interaction. As predicted,
MAP1B-EE neurons had significantly more MAP1B-LC3 PLA puncta
compared to control neurons (Fig. 6e, f, Supplementary Fig. 14a).
Reciprocally, the number of ATG7-LC3 PLA puncta was significantly
reduced in MAP1B-EE neurons compared to control neurons (Fig. 6g, h,
Supplementary Fig. 14b). Therefore, MAP1B-EE sequesters cellular
LC3-I and prevents its lipidation, leading to reduced autophagosome
formation and impaired autophagy in neurons (Fig. 6a).

### Activation of autophagy rescues MAP1B-EE-induced neuronal deficits

Our data show that MAP1B-EE impairs autophagy in neurons which can
be rescued by acute MAP1B knockdown. We next determined whether
activation of autophagy may rescue those deficits we observed in
MAP1B-EE neurons. We first treated mouse MAP1B-EE neurons with
rapamycin, a potent activator of autophagy[61]. Rapamycin treatment
rescued the morphology deficits in MAP1B-EE mouse primary neurons
without significant effect on control neurons (Supplementary Fig. 15).

We then investigated whether rapamycin could rescue the mor-
phological and electrophysiological deficits in 5q13.2trip ASD patient

neurons. Indeed, rapamycin treatment of ASD patient neurons led to
increased dendritic complexity and total dendritic length compared
with vehicle-treated ASD neurons (Fig. 7a–d, Supplementary
Fig. 16a–c). To test the rescue effects of rapamycin treatment on
autophagy activity, we assessed p62 accumulation as well as the
autophagy flux in ASD neurons with or without rapamycin treatment.
As expected, rapamycin treatment reduced p62 accumulation (Sup-
plementary Fig. 16d, e) and increased autophagic vacuoles (Supple-
mentary Fig. 16f, g) in ASD neurons. In addition, MEA recording
showed that rapamycin treatment significantly decreased the mean
firing rate in ASD patient neurons (Fig. 7e–g), similar to what we have
found in ASD patient neurons with MAP1B knockdown (Fig. 2i). Con-
versely, rapamycin treatment did not affect bursting activities and
synchrony index of patient neurons (Fig. 7h, Supplementary
Fig. 16h–j). The reduction of excitability in patient neurons treated
with rapamycin was also confirmed by whole-cell patch-clamp
recording (Fig. 7i, j). Rapamycin-treated patient neurons had sig-
nificantly increased frequency (Fig. 7k, l) and amplitude (Fig. 7m) of
mEPSCs. Therefore, activation of autophagy rescued both morpholo-
gical and electrophysiological deficits in patient iPSC-derived neurons.
Taken together, rapamycin treatment rescues deficits of neurons with
MAP1B-EE.

### Activation of autophagy rescues FMRP deficiency-induced neuronal deficits

Finally, we assessed the effects of either MAP1B genetic reduction or
rapamycin on neurons with FMRP deficiency. Using a similar strategy
as illustrated in Fig. 2c, we found that knocking down *MAP1B* in FXS
patient iPSCs differentiated neurons resulted in a significant increase
in both complexity and total dendritic length, compared with controls
(Fig. 8a, b, Supplementary Fig. 17a, b). In addition, since FXS neurons
exhibited increased excitability (Fig. 8c, Supplementary Fig. S17c–e),
we next assessed the effect of either *MAP1B* knockdown or rapamycin
treatment on the electrical activities of FXS neurons. We found that
knocking down MAP1B or rapamycin treatment significantly reduced
the mean firing rate in FXS2 and burst frequency in FXS1 neurons
(Fig. 8d, Supplementary Fig. 17f–h). Furthermore, MAP1B knockdown
and rapamycin treatment led to significantly increased autophagy
vacuole numbers in both FXS1 and FXS2 neurons (Supplementary
Fig. 17i–l).

We then investigated the effect of rapamycin treatment on the LV-
*shFMR1* infected human mid-fetal cortical tissue slides. A 24-h rapa-
mycin treatment led to increased LC3 levels (Fig. 8e, f) and reduced
p62 accumulation (Supplementary Fig. 18a, b) in both FMRP-deficient
(LV-*shFMR1*) and control (LV-*shNC*) neurons. Noticeably, rapamycin
treatment partially rescued the impaired dendritic morphology of
FMRP-deficient neurons (Fig. 8g–i). These data demonstrate that both
knockdown of MAP1B and activation of autophagy can rescue FMRP
deficiency-induced neuronal deficits.

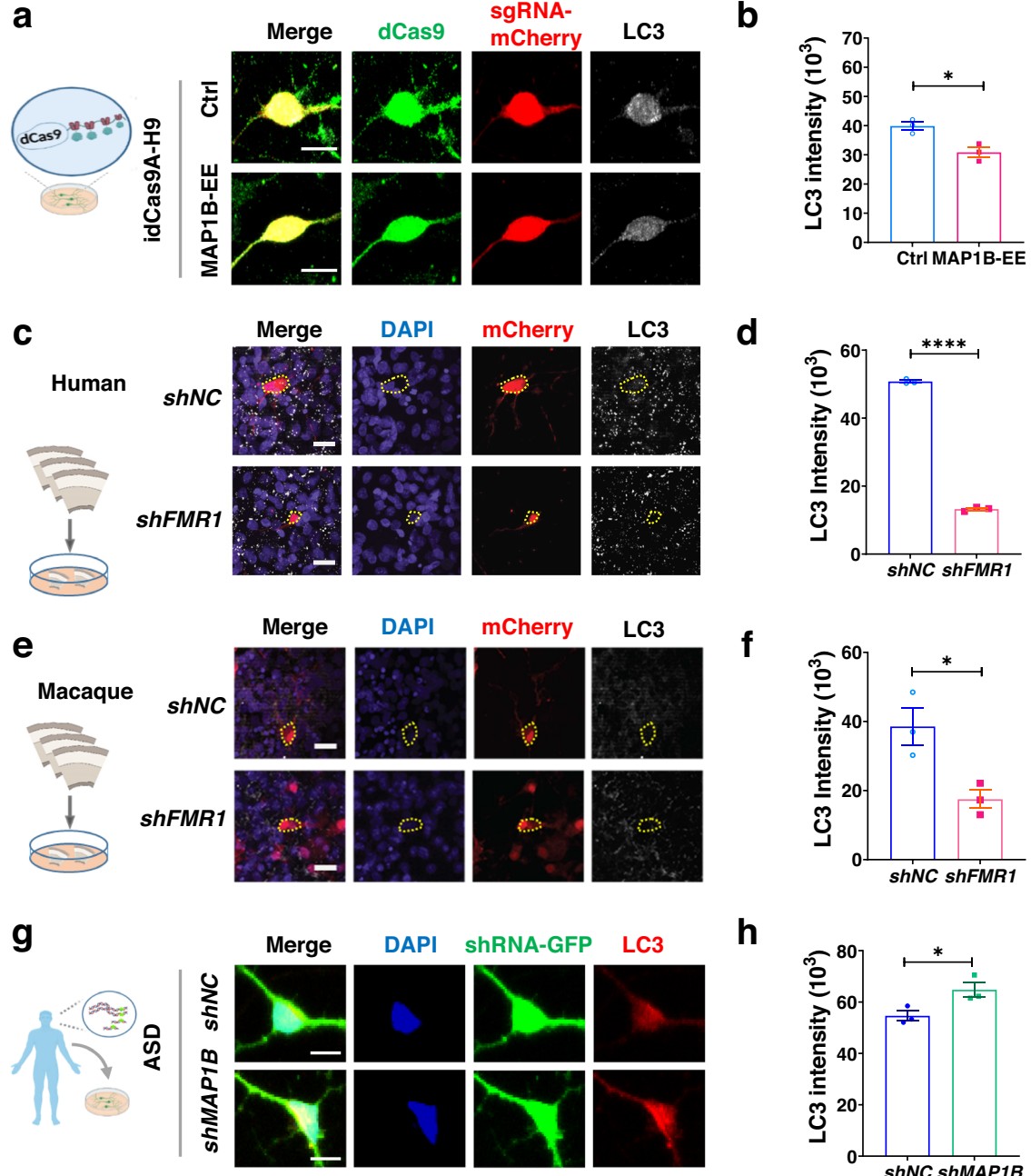

**Fig. 5 | Elevated MAP1B leads to autophagy deficits in human and macaque neurons. a**, **b** Analysis of LC3 intensity of human neurons with MAP1B-EE. Representative confocal images of human neurons (**a**) stained with dCas9-GFP (green), sgRNA-mCherry (red), and LC3 (white). Scale bars, 10 μm. **b** Two-tailed, unpaired Student's *t*-test, *p* = 0.0155. *n* = 3 independent neuronal differentiations, *N* = 1. **c** Representative confocal images of neurons in ex vivo human cortical slices infected with LV-shRNA-mCherry (red) and stained with DAPI (blue) and LC3 (white). Scale bars: 20 μm. **d** Quantification of LC3 intensity in mCherry+ neurons in human cortical slices. Two-tailed, unpaired Student's *t*-test, *p* < 0.0001. *N* = 3 individual cortices. **e** Representative confocal images of neurons in the rhesus macaque cortex expressing shRNA-mCherry (red), LC3 (white) in lentivirus-infected ex vivo macaque cortical slices. Scale bars: 20 μm. **f** Quantification of LC3 intensity in mCherry+ neurons in macaque cortical slices. Two-tailed, unpaired Student's *t*-test, *p* = 0.0238. *N* = 3 individual cortices. **g**, **h** Analysis of LC3 intensity of 5q13.2trip ASD neurons with or without MAP1B knockdown. Representative confocal images (**g**) of neurons stained with DAPI (blue), shRNA-GFP (green) and LC3 (red). Scale bars, 10 μm. **h** Two-tailed, unpaired Student's *t*-test, *p* = 0.0434. *n* = 3 independent neuronal differentiation, *N* = 1. Data are presented as mean ± s.e.m. Source data are provided as a Source Data file.

Collectively, by using a combination of patient-derived and gene-edited hPSCs, human and macaque ex vivo brain slices, and mouse models, our results illustrate that pathologically elevated MAP1B impairs neuronal morphological and physiological development which can be corrected by either genetic reduction of MAP1B or rapamycin treatment to activate autophagy.

## Discussion

Our study demonstrates that not only FMRP controls MAP1B levels in neurons but also such regulation is conserved in primates. Using a combination of experimental systems and methods, we show that elevated expression of MAP1B is detrimental to neuronal development and likely contributes to social behavioral deficits commonly

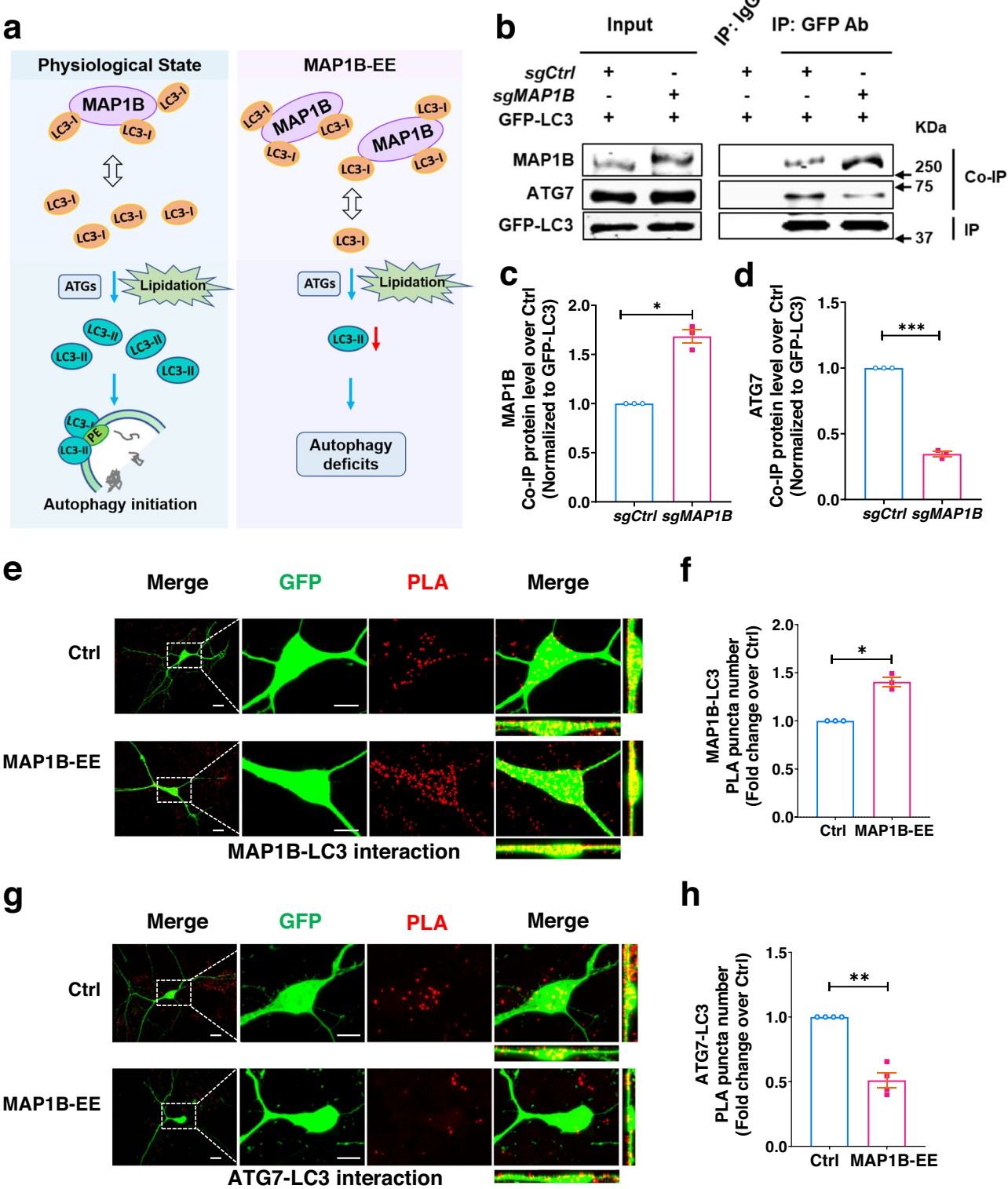

found among FXS and ASD individuals. Therefore, our study unveils a causal link between elevated MAP1B levels and neuronal pathology associated with FXS and ASD. In addition, we show that both genetic reduction of MAP1B and activation of autophagy can correct deficits of 5q13.2trip ASD and FMRP deficient neurons with elevated MAP1B levels, which provides a mechanism for potential therapeutic treatment. Finally, we provide experimental evidence for the impact of FMRP deficiency in human and non-human primate tissues and establish a much-needed experimental model for

studying *FMR1* and other gene functions in primate brain development.

MAP1B has been known as a target of FMRP for nearly 30 years, first identified in *Drosophila*[7] and mouse brains[62], and subsequently in genome-wide FMRP target identification studies of brain tissues and neurons, including human brains and neurons[30,31,63–65]; however, the levels of MAP1B in these studies were not assessed. One study shows that MAP1B protein levels are upregulated in *Fmr1* knockout mouse brains[19]. However, MAP1B protein levels in human FXS brains and

**Fig. 6 | Elevated MAP1B sequesters LC3 and prevents its lipidation in neurons.**
**a** A schematic model showing that MAP1B-EE sequesters LC3-I and prevents its lipidation in neurons, leading to autophagy deficits. **b** Western blot analysis of HEK293T cells co-transfected with GFP-LC3, dCas9 Activator system, and either *sgCtrl* or *sgMAP1B*. MAP1B or ATG proteins interacted with GFP-LC3 were pulled down using an anti-GFP antibody and immunoprecipitation (IP) followed by detection using antibodies against MAP1B, and ATG7. Input, 1% of the total lysate. The samples derive from the same experiment and the gels/blots were processed in parallel. For quantification shown in c and d, protein amounts were normalized to GFP-LC3 and subsequently compared to control cells. **c** Quantitative analysis of MAP1B. Two-tailed unpaired Student's *t*-test with unequal variances, $p = 0.0104$; $n = 3$ biologically independent transfections, $N = 1$. **d** Quantitative analysis of ATG7.

Two-tailed unpaired Student's *t*-test with unequal variances, $p = 0.0008$. $n = 3$ biologically independent transfections, $N = 1$. **e** Proximal Ligation Assay (PLA) analysis showing increased interaction between MAP1B and LC3 in MAP1B-EE mouse primary hippocampal neurons. The PLA signal (red) indicates close proximity of the antigens. Scale bar, 10 μm. **f** Quantification of PLA puncta showing MAP1B and LC3 interaction. $p = 0.0134$. Ctrl: $N = 3$ biological replicates. **g** PLA analysis showing reduced interaction between LC3 and ATG7 in MAP1B-EE mouse primary hippocampal neurons. Scale bar, 10 μm. **h** Quantification of PLA puncta showing LC3 and ATG7 interaction. $p = 0.0034$. $N = 3$ biological replicates. Quantifications in (**f**) and (**h**) were performed after 3D reconstruction and subsequently compared to control cells, two-tailed unpaired Student's *t*-tests with unequal variances were used. All data are presented as mean ± s.e.m. Source data are provided as a Source Data file.

neurons have remained uncharacterized and thus the role of MAP1B in the pathology of FXS has remained unclear. Our study has addressed this long-standing question by illustrating that MAP1B protein levels are indeed higher in FXS human neurons and demonstrating that elevated levels of MAP1B have deleterious impacts on neuronal development.

MAP1B is implicated in the crosstalk between microtubules and actin filaments which is critical for microtubule assembly and stability[9] and is important in neuronal dendritic development, neuronal migration, and axonal guidance[10,12,13]. Loss of function of MAP1B is detrimental in both humans[15] and mice[16,17], and therefore most studies have focused on the consequences of MAP1B mutation and deletion. As expected, the reduction of MAP1B in neurons leads to impaired dendritic development[17,66]. On the other hand, less is known about the pathogenic effect of elevated MAP1B levels. MAP1B protein levels are elevated in brain and spinal neurons lacking Gigaxonin, a protein lost in giant axonal neuropathy[20]. These observations suggest that MAP1B levels must be maintained for proper neuronal development and function. Most studies assessing the impact of elevated MAP1B levels have used methods yielding high levels of MAP1B expression[25–27], which greatly exceed the elevated MAP1B levels found in disease conditions[20] or gene duplication/triplication[24]. In our study, we used a sgRNA-guided dCas9-Activator system to induce (2- to 3-fold) elevated expression of MAP1B (MAP1B-EE) in both mouse and human models, a method that more closely recapitulates the expression levels observed in the context of gene duplications or triplications. Our results provide evidence that MAP1B-EE is pathogenic.

ASD is characterized by impaired social interactions, disrupted communication, and repetitive behaviors. ASD affects one in 36 8 years old children in the United States (CDC 2020 data) and the prevalence of ASD diagnoses has nearly tripled over the last 20 years[67]. A major challenge in understanding ASD and developing effective treatments is deciphering the impact of the diverse genetic changes found in ASD. To date, more than 1000 genes and over 2000 copy number variants (CNVs) have been reported in individuals with ASD[68]. However, most ASD research has focused on a few syndromic ASD genes, and the pathogenic contributions of specific genes affected by ASD-associated CNVs are largely uncharacterized. CNVs involving both deletions[69] and duplications[21] in the 5q13.2 region have been associated with human diseases, including spinal muscular atrophy[70] and alcohol dependence[21]. However, most reported CNVs in 5q13.2 encompass large genomic regions[23] and a comprehensive assessment of the genes affected by 5q13.2 CNVs for their roles in disease pathology has not been performed. Duplication in 5q13.2 has been found in a child with developmental delay and facial dysmorphism[22]. A number of 5q13.2 CNVs have been reported in DECIPHER database (Supplementary Data 2)[24]. However, the present study directly links a 5q13.2 CNV with autism. Our results demonstrate that studying CNVs may provide important and much-needed information about the pathogenesis of ASD. In this study, we assessed four genes affected by a small triplication in 5q13.2 found in an ASD patient and demonstrated that pathogenically elevated MAP1B leads to autophagy deficits and

impairs morphological and physiological development of both mouse and human neurons. We showed that targeted activation of the *Map1b* gene in the PFC impaired social behaviors in mice, unveiling a causal link between MAP1B upregulation and ASD-associated behavioral deficits. Consistent with our findings in mice, we have discovered that a subset of human ASD populations exhibits elevated *MAP1B* mRNA levels in the PFC, suggesting a shared molecular signature and potential therapeutic targets among a subset of ASD patients.

Autophagy mediates the degradation and recycling of cellular components through the lysosomal system, which is critical for neuronal development[51]. It remains unclear which ASD-associated gene mutations or CNVs contribute to autophagy deficits and their underlying mechanisms. Here we have identified the mechanism underlying how elevated MAP1B regulates autophagosome formation and further leads to autophagy deficits in mammalian neurons. Based on our findings, we propose a model to explain how elevated MAP1B impairs autophagosome formation (Supplementary Fig. 19). In this model, there is an equilibrium between MAP1B-bonded LC3-I and cytosolic LC3-I under the physiological state. The cytosolic LC3-I is conjugated to phospholipids to form its lipidated counterpart LC3-II and initiates autophagosome formation. In neurons with MAP1B-EE, MAP1B competitively binds LC3-I and blocks its interaction with ATG7, which results in less autophagosome formation and impaired autophagy. Autophagy is a conserved and essential cellular pathway and is known to play important roles in many essential processes in the brain, including neurogenesis, neuronal maturation, synaptic pruning, and neural plasticity[51]. Downregulated autophagy has been observed in postmortem brain tissues of ASD patients and several ASD mouse models[52,54,71]. Activation of mTOR signaling leads to reduced LC3 lipidation to generate LC3-II, resulting in reduced autophagosome formation[51]. Mice deficient in tuberous sclerosis 2 (TSC2), an inhibitor of mTOR, exhibit constitutive activation of mTOR and impaired autophagy, which can be corrected by rapamycin treatment[52]. In our study, we did not detect mTOR dysregulation in MAP1B-EE neurons, but rapamycin treatment was capable of rescuing the morphology deficits caused by MAP1B-EE in mouse neurons. We reasoned that both mTOR activation and MAP1B-EE impact LC3 lipidation, but through independent mechanisms, therefore inhibition of mTOR signaling by rapamycin could "correct" or "rebalance" the reduction of LC3 lipidation caused by MAP1B-EE. In fact, we demonstrate that both the genetic reduction of MAP1B by shRNA knockdown and the treatment with rapamycin could rescue morphological and electrophysiological deficits of FXS and 5q13.2trip ASD patient-derived neurons.

Elevated excitability of FMRP-deficient developing neurons has been shown in a number of experimental models for FXS[42,72,73]. For example, elevated cortical excitability has been found in the neocortex[74–76] and hippocampal CA1[77–79] of *Fmr1* knockout mice. Altered excitability has also been observed in an *Fmr1* knockout rat model[80,81]. In addition, human neurons differentiated from FXS iPSCs exhibit increased excitability and can be rescued by reactivation of the *FMR1* gene[44]. Therefore, hyperexcitability seems to be a shared

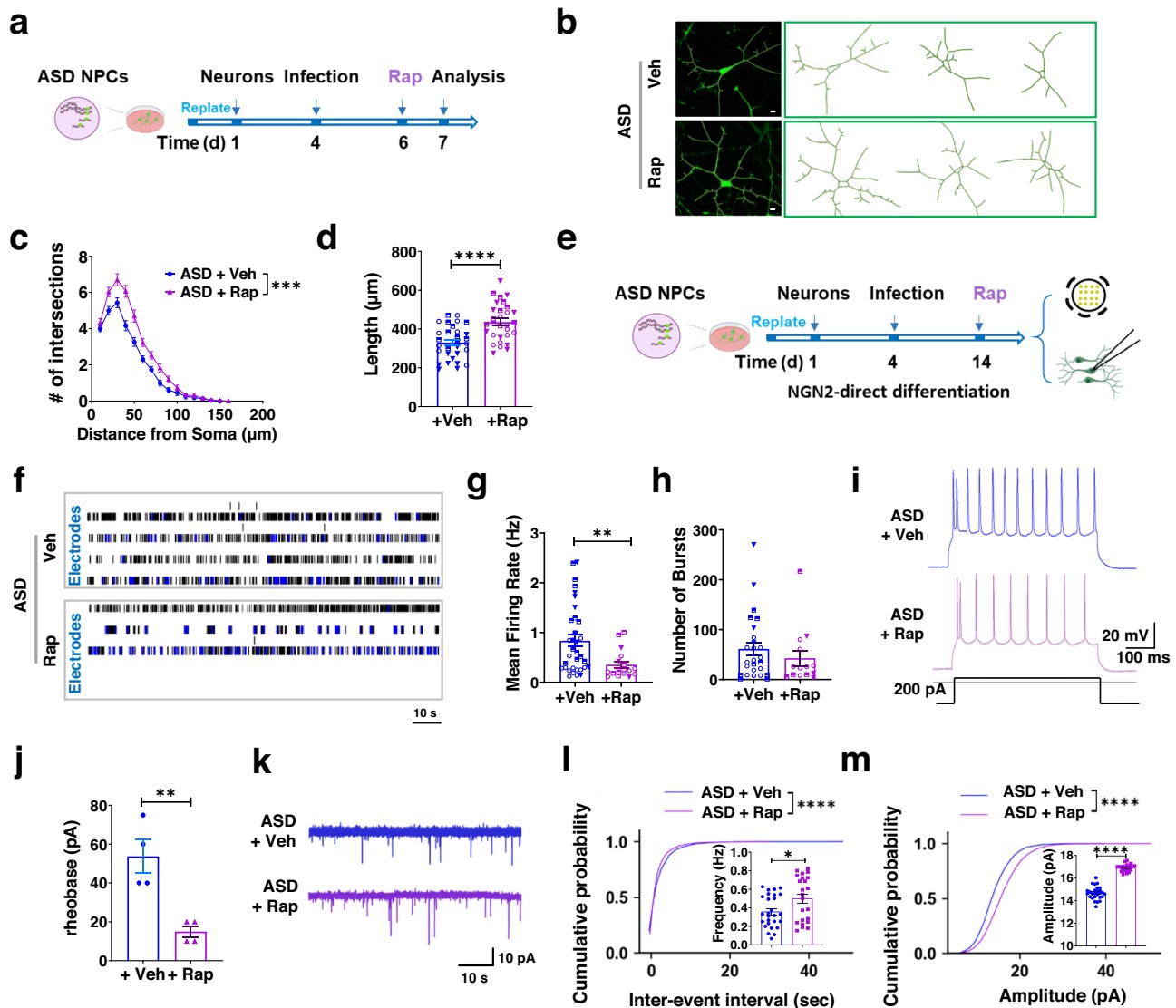

**Fig. 7 | Activation of autophagy rescues MAP1B-EE-induced neuronal deficits in human neurons. a, b** Experimental scheme (**a**) and representative confocal images (from three independent experiments) and traces of GFP+ neurons (**b**) for assessing the effect of rapamycin (Rap) on 5q13.2trip ASD neurons. Scale bar, 10 μm. **c** Sholl analysis of ASD patient neurons treated with DMSO (Vehicle, Veh) or rapamycin (Rap, 300 nM). MANOVA, $F_{(1,58)} = 27.747$, $p < 0.001$. $n = 30$ cells from three independent neuronal differentiations, $N = 1$. **d** Total dendritic length. Two-tailed, unpaired Student's $t$-test, $p < 0.0001$. $n = 30$ neurons from three independent neuronal differentiations, $N = 1$. **e, f** Experimental scheme (**e**) and representative raster plots (**f**) for accessing the electrophysiology of ASD patient neurons treated with rapamycin. **g** Quantifications of neuronal mean firing rate. Two-tailed, unpaired Student's $t$-test, $p = 0.0081$. ASD Veh: $n = 34$ individual wells, ASD Rap: $n = 17$ individual wells from three individual neuronal differentiations, $N = 1$. **h** Number of bursts. Two-tailed, unpaired Student's $t$-test, $p = 0.3723$. Only the wells with bursting activity were analyzed for burst-related parameters, ASD Veh: $n = 26$ individual wells; ASD Rap: $n = 14$ individual wells from three independent neuronal differentiations, $N = 1$. **i** Representative traces showing current injection-evoked action potentials recorded from patient neurons with or without rapamycin treatment. **j** Rheobase current threshold of ASD neurons with rapamycin treatment. Two-tailed, unpaired Student's $t$-test, $p = 0.0050$. ASD Veh: $n = 4$ neurons, ASD Rap: $n = 4$ neurons from three independent neuronal differentiations, $N = 1$. **k** Representative traces of mEPSCs recorded from patient neurons with or without rapamycin treatment. **l, m** Cumulative probability of inter-event intervals and amplitudes of mEPSCs. Mann–Whitney Rank Sum test, $p < 0.0001$. (Inset: mean mEPSC frequency, $p = 0.0229$ and amplitude, $p < 0.0001$. Two-sided, unpaired Student's $t$-test). ASD Veh: $n = 27$ neurons, ASD Rap: $n = 23$ neurons from three independent neuronal differentiations, $N = 1$. All error bars reflect mean ± s.e.m. Source data are provided as a Source Data file.

phenotype of developing neurons resulting from FMRP deficiency across species. Our study suggests that targeting downstream effectors, such as MAP1B, or deficient pathways, such as treatment with rapamycin, may be potential methods for reversing the deficits of neuronal excitability in FXS.

In summary, our study has established a causal link between MAP1B elevation and ASD phenotypes, revealing a mechanism and a potential treatment targeting the autophagy pathway. Many genes have been implicated in the development of ASD. Although *MAP1B*

gene copy number gain has only been found in a limited number of patients with ASD diagnosis (Supplementary Data 2), we found that among PFC samples collected by PsychENCODE, the ASD population has ~1.5-fold elevated *MAP1B* mRNA levels compared to controls. Whether elevated *MAP1B* mRNA levels are a cause or consequence, mildly elevated MAP1B levels are indeed associated with a subset of ASD cases in humans. Targeting the autophagy pathway, such as using rapamycin treatment, might be a promising potential therapeutic treatment for FXS and ASD.

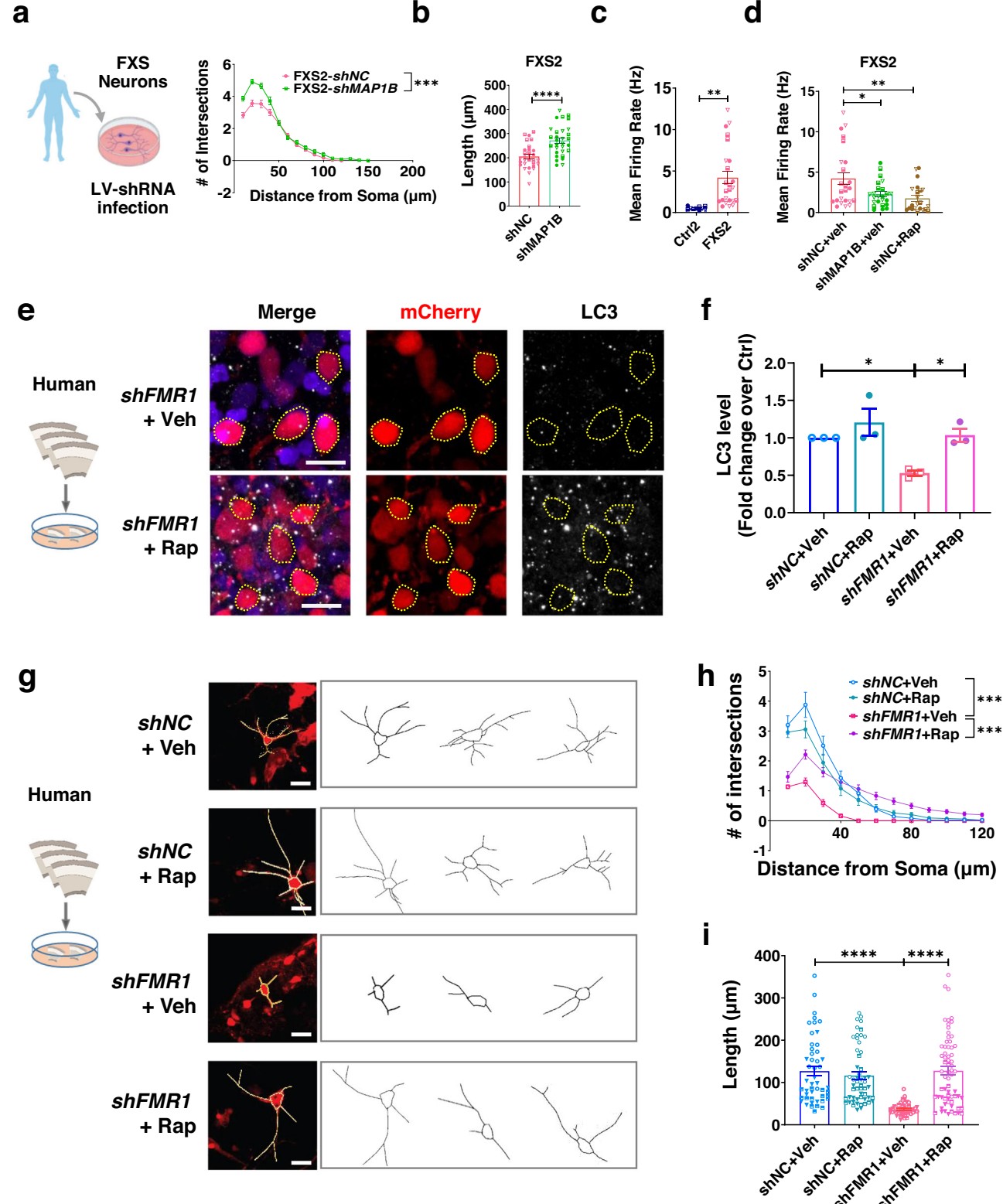

## Methods

### Mice and animal husbandry

We performed all procedures involving live mice (*Mus musculus*) in accordance with the NIH Guide for the Care and Use of Laboratory Animals and the protocols approved by the University of Wisconsin-Madison Animal Care and Use Committee (protocol number G005741 for mice; G006583 for macaque). The University of Wisconsin holds an NIH Public Health Animal Welfare Assurance (ID number A3368-01). The

C57BL/6J mice (JAX stock #000664) and dCas9Activator mice[37] (JAX stock #031645) were purchased from Jackson Laboratory and maintained in house[82]. All the mice have been bred onto a C57BL/6J genetic background. Both male and female neonatal mice (postnatal day 0-P2) were used for neuronal isolation. Young adult (10–12 weeks old) male mice were used for behavior tests. Mice were housed in groups and maintained on a 12-h light/dark cycle with food and water available ad libitum. The experiments were conducted during the light phase.

**Fig. 8 | Activation of autophagy rescues FMRP deficiency-induced neuronal deficits. a** Sholl analysis of FXS2 neurons with MAP1B knockdown. $F_{(1,63)} = 20.697$, $p < 0.001$, *shNC* $n = 30$ neurons, *shMAP1B* $n = 35$ neurons from three independent neuronal differentiations, $N = 1$. **b** Quantification of total dendritic length. Two-tailed, unpaired Student's *t*-test, $p < 0.0001$, *shNC* $n = 30$ neurons, *shMAP1B* $n = 35$ neurons from three independent neuronal differentiations, $N = 1$. **c** Neuronal mean firing rate. Two-tailed, unpaired Student's *t*-test, $p = 0.0032$. Ctrl2: $n = 10$ individual wells, FXS2: $n = 24$ individual wells from three independent neuronal differentiations, $N = 1$. **d** Mean firing rate in FXS2 neurons with MAP1B knockdown or rapamycin treatment. One-way ANOVA with Dunnett post hoc tests, *shNC*+Veh vs. *shMAP1B*+Veh: $p = 0.0165$; *shNC*+Veh vs. *shNC*+Rap: $p = 0.0027$. *shNC* Veh: $n = 24$ individual wells, *shMAP1B* Veh: $n = 24$ individual wells, *shNC* Rap: $n = 22$ wells from three independent neuronal differentiations, $N = 1$. **e** Representative confocal images of neurons expressing shRNA-mCherry (red), LC3 (white) in lentivirus-infected ex vivo human cortical slices. Scale bars: 20 μm. **f** LC3 intensity in mCherry + neurons in human cortical slices. Two-way ANOVA with two-sided Bonferroni post hoc analysis for multiple comparisons: *shNC*+Veh vs. *shFMR1*+Veh: $p = 0.0476$; *shFMR1*+Veh vs. *shFMR1*+Rap: $p = 0.0332$. $N = 3$ individual cortices. **g** Representative confocal images (from three independent experiments) and Neurolucida traces of mCherry+ neurons. Scale bar, 10 μm. **h** Sholl analysis. MANOVA, *shNC*+Veh vs. *shFMR1* Veh: $F_{(1,91)} = 51.474$, $p < 0.001$; *shFMR1*+Veh vs. *shFMR1* Rap: $F_{(1,108)} = 54.811$, $p < 0.001$. **i** Total dendritic length of LV-*shFMR1* or LV-*shNC*-infected neurons treated with Veh or Rap. Two-way ANOVA with two-sided Bonferroni post hoc analysis for multiple comparisons, *shNC*+Veh vs. *shFMR1*+Veh: $p < 0.0001$; *shFMR1*+Veh vs. *shFMR1*+Rap: $p < 0.0001$. For all data shown in (**g**, **h**), $n = 49$ (*shNC* +Veh) neurons; $n = 54$ neurons (*shNC*+Rap); $n = 44$ neurons (*shFMR1*+Veh); $n = 66$ neurons (*shFMR1*+Rap) from $N = 3$ individual cortices. All error bars reflect mean ± s.e.m. Source data are provided as a Source Data file.

## Data collection timing and blinding
Data collection was carried out for a predetermined period of time, as dictated by literature or core facility-based standards. All cell counting, tracing, quantification, and behavioral analyses were performed by experimenters who were blind to the identity and treatments of the samples.

## Tissue procurement
All work was performed according to NIH guidelines for the acquisition and distribution of human tissue for bio-medical research purposes. Human (*Homo sapiens*) fetal tissue was collected by the Birth Defects Research Laboratory (BDRL) at the University of Washington in compliance with all Federal, State, and Institutional regulations. Donors are recruited at clinical sites in the Puget Sound area under an approved human subjects research protocol at the University of Washington. After a patient has consented in writing to pregnancy termination with a clinical provider, she is then consented to research participation by a second staff member trained in human subjects research and HIPAA regulations. The research consent form is signed by the participant and the second staff member. Research participation does not affect the method of termination, nor are enticements of any type provided to the participant, clinic staff, or research team members. We used de-identified postmortem human brain specimens (115, 115, 137 PCD) This study was performed in accordance with ethical and legal guidelines of the University of Wisconsin-Madison Institutional Review Board. Appropriate informed consent was obtained and all available non-identifying information was recorded for each specimen. Tissue was handled in accordance with ethical guidelines and regulations for the research use of human brain tissue set forth by the NIH (http://bioethics. od.nih.gov/humantissue.html) and the WMA Declaration of Helsinki (http://www.wma.net/en/30publications/10policies/b3/index.html).

All experiments using non-human primates Rhesus macaque (*Macaca mulatta*) (91, 101, and 104 PCD) were carried out in accordance with a protocol approved by the University of Wisconsin's Institutional Animal Care and Use Committee and NIH guidelines. All clinical histories, tissue specimens, and histological sections were evaluated to assess for signs of disease, injury, and gross anatomical and histological alterations. No obvious signs of neuropathological alterations were observed in any of the macaque specimens analyzed in this study.

## Ex vivo fetal organotypic slice cultures, viral infection, and chemical treatment
Cortical walls from human and macaque brains at midgestation were dissected in Hibernate E solution (Thermo Fisher, #A1247601) supplemented with 1X B27 (Thermo Fisher 17504044), 1X Glutamax (Thermo Fisher 35050061), and 1 X Penicillin-Streptomycin (Thermo Fisher 15140122). Tissue was then embedded in 3.2% low melting point agarose (American Bioanalytical) in PBS and sectioned in a Leica VT1200 S Vibrating blade microtome. Following this, they were transferred into ice-chilled artificial cerebrospinal fluid (ACSF; 125 mM NaCl, 2.5 mM KCl, 1.25 mM NaH$_2$PO$_4$, 1.0 mM MgCl$_2$, 2.0 mM CaCl$_2$, 25 mM NaHCO$_3$, 25 mM dextrose, bubbled with 95% O$_2$/5% CO$_2$). Then, 300-μm slices were transferred to 0.4-μm pore size, PET track-etched membrane, cell culture inserts (Falcon #353090; #353180) in Brain-PhysTM Neuronal Medium (Stem Cell Technologies, #05790) supplemented with N$_2$ (1:100), BDNF (25 ng/ml), and B27 (1:50) as described[83].

Next, 10 μl lentivirus (LV-*shFMR1*-mCherry or LV-*shNC*-mCherry) was added to 500 μl of culture medium per well. At DIV10, 300 nM Rapamycin[84] (LC Laboratories, Cat# R-5000) or DMSO as vehicle was administered for 24 h, respectively, prior to fixation for analysis.

## Histological and neuronal morphology analysis of cultured fetal organotypic slices
Fetal organotypic brain slices were fixed in fixative (Sakura, Tissue-Tek VIP Fixative) for 24 h. Organotypic slices were cryoprotected in sucrose 30%, embedded in OCT, and sectioned using cryostat at 40 μm. For immunostaining, tissue sections were pre-blocked with TBS++ (TBS containing 3% goat or donkey serum and 0.2% Triton X-100) for 1 h at room temperature, followed by incubation with primary antibodies diluted in TBS++ overnight at 4 °C. After washing three times, secondary antibodies were incubated 1 h at room temperature. All sections were counterstained with a nuclear counter stain, DAPI (4′, 6-diamidine-2′-phenylindole dihydrochloride, 1:2000, Roche Applied Science, Indianapolis, IN). After staining, autofluorescence Eliminator Reagent (Millipore, 2160) was used to reduce autofluorescence according to the manufacturer's instructions. Sections were mounted and maintained at −20 °C in the dark until analysis. The signal intensity of LC3 and MAP1B of mCherry positive cells was quantified using Image J software as previously described[39]. The z-stack images (1 μm interval) were acquired using a Nikon A1 confocal microscope. At least 50 mCherry positive cells were randomly selected from 3 cortical sections in each individual sample and the fluorescent intensity of LC3 or MAP1B was measured after subtracting background pixel intensity in the same image using Image J software (NIH). The dendritic morphology of a single neuron was analyzed by Neurolucida software with a 3D module plug-in (MicroBrightField, Inc. Williston, VI, http://www.mbfbioscience. com/). More than 30 neurons were analyzed for each group in each experiment. Samples from three individual biological samples were analyzed for experimental conditions except as specified.

## Microarray analysis of human fibroblasts and NPCs
Isolated genomic DNA was quantified, amplified, fragmented, and hybridized to the Illumina CytoSNP850K bead chip that contains 850,000 different locus-specific 50-mer probes with at least 15x redundancies. The 850,000 probes have an average probe spacing of 1.8-kilobases (kb) across the whole genome (backbone coverage) and increased probe spacing (1-kb) in targeted cytogenetically relevant genes. The minimal resolution of a CNV is 36 kb across the genome and 20 kb in targeted regions. Fluorescence type and intensity of each probe were analyzed by Illumina's Genome Studio

and BlueFuse software. Data analysis was performed using BlueFuse v4.4 and the GRCh37/hg19 human genome assembly from February 2009. Other databases include the following: UCSC Genome Browser (http://genome.ucsc.edu/), ClinGen (https://www.ncbi.nlm.nih.gov/projects/dbvar/clingen/), Database of Genomic Variants (http://dgv.tcag.ca/dgv/app/home), and DECIPHER (https://decipher.sanger.ac.uk/). Copy number variant classification follows ACMG guidelines[85].

### Gene expression analysis in humans

*MAP1B* gene expression levels are the transcripts per million (TPM) values of transcriptomic (RNA-seq) analyses of the DLPFC region (BA9) of postmortem brain tissue collected by the PsychENCODE project[49]. The project includes 1166 controls (CTR), 31 ASD, 172 bipolar disorder (BPD), and 497 schizophrenia (SCZ) patients. The one-side *t*-test was performed to test the significance of upregulation in ASD compared to CTR.

### Plasmids

Please see Supplementary Data 3. LV-*shMAP1B*, *shMap1b*, *shZnf366*, and *shNC* were cloned using LV-*shNC* vector[39,86] as a backbone and the U6 or H1-shRNA cassettes were also cloned into the backbone through HpaI/ClaI restriction sites. The efficiency and specificity of shRNA knockdown were determined by transfecting into Neuro2A or HEK293T cells using Lipofectamine 2000 (Invitrogen, #11668-027), and analyzed at 48 h post-transfection by qPCR.

LV-dCAS-VP64 (Addgene #61425), LV-MS2-P65-HSF1 (Addgene #61426), and LV-sgRNA (Addgene #61427) (SAM system) were created by F. Zhang[48]. Five sgRNAs targeting the proximal promoter of mouse *Map1b* gene (*sgMap1b* #1–5), between −305 bp and −24 bp relative to the TSS, were designed in Benchling based on published scoring methods[87] (sequences are included in Supplementary Data 3) and cloned to the same lentiviral sgRNA vector (Addgene #61427). The efficiency of *Map1b* activation was determined by transfecting each sgRNA together with the dCas9-VP64 and SAM vectors into Neuro2A cells using Lipofectamine followed by qPCR analysis at 48 h post-transfection. Five sgRNAs targeting the proximal promoter of the human *MAP1B* gene (*sgMAP1B* #1–5), between −101 bp and +182 bp relative to the TSS, were designed and cloned using the same method. The efficiency of *MAP1B* activation was determined by transfecting each sgRNA together with the dCas9-VP64 and SAM vectors into HEK293T cells.

A hSyn1-flex mCherry cassette was inserted between restriction sites KpnI and BglII of PX552 (Addgene #60958)[88] to create PX552-hSyn1-flex-mcherry plasmid[82]. The *sgMap1b* #2 was selected to clone to the PX552 vector following the protocol provided by Addgene.

Plasmid 49535-*sgAAVS1*-T2 was cloned by inserting the *sgAAVS1*-T2 sequence (Addgene #41818) to a Cas9-expression vector (Addgene #49535) using published methods[33]. Donor plasmid AAVS1-TRE3G-dCas9-10xGCN was cloned by inserting the dCas9-10xGCN-GFP sequence (from Addgene #107307)[37] to AAVS1-TRE3G-EGFP plasmid (Addgene #52343) through MluI/SalI restriction sites.

FUdeltaGW-rtTA (Addgene #19780), pLV-TetO-hNGN2-Neo (Addgene #99378), and pBABEpuro-GFP-LC3 (Addgene #22405) plasmids were obtained from Addgene. pCAG-mCherry/GFP/LC3-IRES-Cre was a gift from Dr. D. Chichung Lie[56].

### Real-time quantitative PCR (qPCR)

Real-time PCR assays were performed using standard methods as previously described[39,89]. The first-strand cDNA was generated by reverse transcription with both oligo dT primer and random primers using PrimeScript™ RT Reagent Kit (Takara, #RR037A). To quantify the mRNA levels using real-time PCR, aliquots of first-strand cDNA were amplified with gene-specific primers and universal SYBR Green PCR supermix (Bio-Rad, #172-5124) using a Step-1 Real-Time PCR System (Applied

Biosystems). The PCR reactions contained 20–40 ng of cDNA and 300 nM of forward and reverse primers in a final reaction volume of 20 µl. The primers used for PCR are listed in Supplementary Data 3.

### Tissue preparation and immunohistochemistry

Histological analyses of mouse brains were performed using published methods with modifications[82,89]. Briefly, mice were euthanized by intraperitoneal injection of a mixture of 100 mg/kg of Ketamine, 20 mg/kg of xylazine, and 3 mg/kg of acepromazine, followed by transcardial perfusion with saline followed by 4% PFA. Brains were dissected out, post-fixed overnight in 4% PFA, and then equilibrated in 30% sucrose. Then, 40-µm brain sections were generated using a sliding microtome and stored in a −20 °C freezer as floating sections in 96-well plates filled with cryoprotectant solution (glycerol, ethylene glycol, and 0.1 M phosphate buffer, pH 7.4, 1:1:2 by volume). Immunohistology was performed as published previously[82]. The tissue sections were pre-blocked with TBS++ (TBS containing 3% goat or donkey serum and 0.2% Triton X-100) for 1 h at room temperature, followed by incubation with primary antibodies diluted in TBS++ overnight at 4 °C. After washing three times, secondary antibodies were incubated 1 h at room temperature. All sections were counterstained with a nuclear counter stain, DAPI (4′, 6-diamidine-2′-phenylindole dihydrochloride, 1:2000, Roche Applied Science, Indianapolis, IN). After staining, sections were mounted and maintained at −20 °C in the dark until analysis.

### Production of lentivirus

Lentivirus production was performed using published methods[39,89] with modification. Briefly, lentiviral DNA was co-transfected with packaging plasmids pMDL, REV, and pCMV-Vsvg into HEK293T cells using the PEI method. The viral transfer vector DNA and packaging plasmid DNA were transfected into 5 × 15-cm dishes of cultured HEK293T cells using the PEI. The medium containing lentivirus was collected at 48 and 72 h post-transfection, pooled, filtered through a 0.2-µm filter, and concentrated using an ultracentrifuge at 45,000 × *g* for 2 h at 4 °C using an SW32.1 rotor (Beckman). The virus was washed once and then resuspended in 100 µl PBS. We routinely obtained $1 \times 10^9$ infectious viral particles/ml for lentivirus. HEK293T cells used in this study were obtained from Dr. Fred Gage (The Salk Institute).

### Isolation, transfection, and in vitro analysis of mouse primary neurons

Hippocampal and cortical neurons were isolated and transfected using published methods[39,43]. Briefly, WT or dCas9Activator P0 neonate mice were anesthetized using cryo-anesthesia (buried under ice in a plastic bag for 5 min). The hippocampal and cortical neurons were isolated from the hippocampal and cortical tissue of neonate mice and were grown as dispersed cultures. Neurons were transfected with plasmids of interest using Lipofectamine 2000 (Invitrogen, #11668-027) on DIV 4 as they were undergoing dendritic morphogenesis. At 72 h after transfection (DIV 7), transfected neurons were fixed with 4% paraformaldehyde, washed with Dulbecco's phosphate-buffered saline, and coverslipped using DABCO-PVA.

### Morphological analysis of cultured human and mouse neurons

Dendritic morphological analysis was carried out using an Olympus BX51 microscope with a ×20 lens, a motorized stage, and a digital camera as described[39,43,86]. Soma and dendrites of neurons were traced and analyzed using Neurolucida software (MicroBrightField). Samples from at least three individual animals, each from a different litter, were analyzed for experimental conditions for mouse neurons. Samples from at least three independent differentiations were analyzed for experimental conditions for hPSC-derived neurons. Samples from three individual cortices were analyzed for experimental conditions for human and macaque ex vivo slices, only the cells with at least one process were chosen for tracing.

## In vivo AAV grafting followed by behavioral tests

Production of custom AAV8-*sgMap1b*-hSyn1-flex-mcherry and AAV8-*sgCtrl*-hSyn1-flex-mCherry was performed by Vigene Biosciences. AAV9-*pCamK2a*-HIGFP-Cre was purchased from Addgene (Addgene, #105551). Stereotaxic injection of AAV (titers approximately $1 \times 10^{13}$/ml) was performed as described[82]. Briefly, 8- to 10-week-old male mice were anesthetized with isoflurane and placed in a stereotactic instrument (DAVID KOPF Instruments, Tujunga, CA). Microinjections were performed using custom-made injection 33-gauge needles (Hamilton, #776206, Reno, NV, USA) connected to a 10 μl syringe (Hamilton, #87930). AAV (250 nl AAV8-*sgMap1b* or *sgCtrl* together with 250 nl AAV9-*pCamK2a*-HIGFP-Cre) was stereotaxically injected into the bilateral medial prefrontal cortex of adult mice using the following coordinates relative to bregma, caudal: +2.0 mm; lateral: ±0.3 mm; ventral: −2.2 mm, 14° angle pointing lateral. All mice were given at least 21 days to recover before subjecting them to behavioral tests. All behavioral tests started with an open field test to evaluate locomotor activity and general anxiety, in the order described below.

## Open field activity (OF)

This test was performed as previously described to measure locomotive activity[82]. Mice were handled for approximately 15 min a day for a minimum of 7 days prior to the experiment. Before the trial or test phase, mice were brought into the testing room and were allowed to acclimate for at least 60 min. The activity was recorded at 10-min intervals for a total 30-min by means of a computer-operated tracking system (Omnitech Electronics Inc, USA). Total distance moved, and distance traveled in the center and periphery of the arena were measured by the tracking system.

## Elevated plus maze (EPM)

This test was performed following basic procedures to measure anxiety levels as previously described[82]. For testing anxiety, mice were allowed to freely explore the two open arms or two closed (walled) arms on elevated runways 20 cm above the ground for 5 min. More anxious mice prefer closed arms versus open space. All experiments were videotaped and the time in the open arms was scored by ANY-maze (Stoelting).

## Marble burying test (MB)

This test was performed following basic procedures to measure compulsive behaviors[82]. Mice were placed into testing cages (40 × 20 × 30 cm, bedding depth: 3 cm) each containing 20 glass marbles (laid out in four rows of five marbles equidistant from one another). At the end of the 30-min exploration period, mice were carefully removed from the testing cages. All experiments were photographically recorded, and the number of marbles buried was scored by scientists who were blinded to experimental conditions to ensure accuracy. The marble-burying index was arbitrarily defined as the following: 1 for marbles totally covered with bedding, 0.5 for marbles covered >50% with bedding, or 0 for anything less.

## Novel location test (NLT)

This test was performed as previously described to measure spatial learning[82]. Briefly, testing consisted of five 6-min trials, with a 3-min intertrial interval between each trial. In the first trial (Trial 1, Pre-Exposure), each mouse was placed individually into the center of the otherwise empty open arena (38.5 cm long × 38.5 cm wide, and 25.5 cm high walls) for 6 min. For the next three trials (Trials 2–4, Trainings), two identical objects were placed inside, and each mouse was placed individually into the arena and was allowed to explore for 6 min. In the last trial (Trial 5, Test), one of the objects was moved to a novel location, and the mouse was allowed to explore the objects for 6 min. All experiments were videotaped and scored by ANY-maze (Stoelting). The discrimination index was calculated as the percentage of time spent investigating the object in the new location minus the percentage of time spent investigating the object in the old location.

## Three-chamber sociability tests (SI, SN)

Three-chamber social behavior test was carried out as previously described[82]. The social behavior testing apparatus was a 3-chambered box (each chamber: 40 cm long × 22 cm wide, and 23 cm high walls) with clear Plexiglas containing small circular doors (7 cm in diameter) allowing access into each chamber. Briefly, testing consisted of three 10-min sessions: habituation, tests of social interaction/interest (SI), and social novelty recognition (SN). In the habituation session, each mouse was allowed to explore all three chambers freely. In SI, an unfamiliar mouse (stranger 1, age and sex matched to the test mouse) was placed in a round wire cage in one of the compartments, whereas the wire cage in the opposite side compartment contained a motionless toy. The test mouse was tested for the evaluation of social interaction. In SN, one side compartment contained the familiar mouse (stranger 1 from the previous social interaction phase), and the other side contained a novel unfamiliar mouse (stranger 2). The test mouse was tested for evaluation of the social novelty recognition test. All experiments were videotaped and scored by ANY-maze (Stoelting). The discrimination index was calculated as the difference between the percentages of time spent exploring the stranger 1 and toy in Session 2, or stranger 2 and the familiar stranger 1 in Session 3.

## Immunocytochemistry

Immunocytochemical analysis of cultured mouse or human neurons was carried out as described[39,82]. Neurons cultured on coated glass coverslips were fixed with 4% PFA for 15 min at room temperature and then washed twice with PBS. Neurons were blocked with TBS++ (TBS containing 3% goat or donkey serum and 0.2% Triton X-100) for 1 h at room temperature. Primary antibodies were diluted in TBS++ and incubated overnight at 4 °C. Samples were washed three times with TBS and incubated with Alexa Fluor-conjugated secondary antibodies (Thermo Fisher Scientific) diluted in TBS++ for 60 min at room temperature. After three washes with PBS, neurons were counterstained with a nuclear counter stain, DAPI (4′, 6-diamidine-2′-phenylindole dihydrochloride, 1:2000, Roche Applied Science, Indianapolis, IN) and coverslipped in DAVCO-PVA. Confocal images of stained neurons were collected using a Nikon A1 confocal microscope.

## Western blot

Primary antibodies used in this study are listed in Supplementary Data 3. Cells were lysed in RIPA buffer (50 mM Tris, pH 8.0, 150 mM NaCl, 1% NP-40, 0.1% SDS) supplemented with protease inhibitors (Roche Applied Science). After centrifugation for 10 min at 4 °C, supernatants were quantified by Protein Assay Dye Reagent Concentrate (Bio-Rad). 12 μg of total proteins were resolved by SDS-PAGE, transferred to a nitrocellulose membrane, blocked with 5% BSA, and probed with primary antibodies. Secondary antibodies conjugated with near-infrared fluorescent dyes (IRDye 800CW or IRDye 680LT, LI-COR) were used at a dilution of 1:10,000 for visualizing protein bands with an Odyssey Imager (LI-COR). Quantification of intensity was performed using FIJI (Image J) software. At least three independent biological replicates were used for each experiment. Uncropped scans are provided in the Source Data file and supplementary figures.

## Co-immunoprecipitation (Co-IP)

Co-IP followed by Western blot analysis was performed using a published protocol[65]. Briefly, HEK293T cells transfected with sgRNA together with the dCas9 and GFP-LC3 vectors were harvested and homogenized in Lysis buffer (25 mM Tris (pH 7.5), 150 mM NaCl, 1 mM EDTA, 5% Glycerol, 1% IGEPAL) with 1× complete protease inhibitors

(Roche Applied Science). Nuclei and debris were pelleted at $14,000 \times g$ for 10 min. An aliquot of input was saved for WB (50 μl). A monoclonal antibody against GFP was incubated with 1 mg total protein lysate at 4 °C for 4 h and before adding Dynabeads (Life Technologies). The antibody-GFP conjugation was rotated at 4 °C overnight. After the fourth wash with the Wash buffer (25 mM Tris (pH 7.5), 150 mM NaCl, 1 mM EDTA, 1% IGEPAL), the immunoprecipitations were resuspended into 2x SDS-loading buffer. Samples were boiled for 15 min before loading.

## Proximity ligation assays

Cultured mouse primary hippocampal neurons (DIV7) were assessed by PLA according to the Duolink PLA Fluorescence Protocol (Sigma-Aldrich). Briefly, Neurons grown on coverslips were fixed with 4% paraformaldehyde in PBS for 20 min at RT, washed with PBS (3 × 5 min), permeabilized with 0.2% Triton X-100 in PBS for 15 min, and blocked with the Duolink® Blocking Solution for 60 min at RT. The cells were then incubated at 4 °C overnight in Duolink® Antibody Diluent containing the following primary antibodies: rabbit anti-LC3 and mouse anti-MAP1B or rabbit anti-ATG7 and mouse anti-LC3. After washes in 1× Wash Buffer A (Sigma-Aldrich, DUO82049), neurons were incubated with two PLA probes (Duolink In Situ PLA Probe Anti-Rabbit PLUS DUO92002 and Anti-Mouse MINUS DUO92004, Sigma-Aldrich) in Duolink® Antibody Diluent for 1 h at 37 °C, followed by washes with 1x Wash Buffer A and incubation in ligation solution (Duolink In Situ Detection Reagents Red, Sigma-Aldrich Duo92008) for 30 min at 37 °C. Neurons were then washed with Wash Buffer A and incubated in the amplification solution (Sigma-Aldrich) for 100 min at 37 °C to fluorescently label the ligated PLA probes. After amplification, neurons were further incubated with primary antibody against GFP for 60 min at RT, washed in PBST, and incubated with secondary antibody for 60 min at RT. Neurons were then washed in PBS (3 × 5 min), stained with DAPI, and rinsed in Wash Buffer B and 0.01x Wash Buffer B (Sigma-Aldrich, DUO82049). Microscopic imaging (×60) was done with a Nikon A1 confocal microscope.

## Autophagy detection

The autophagy activity in human iPSC-derived neurons was detected with an autophagy detection kit according to the manufacturer's instructions (ab139484, Abcam). Neurons were cultured on coverslips and subjected to mixed dyes of green detection reagent (1:500) and Hoechst 33342 (1:1000) for autophagic vacuoles and nuclei staining, respectively, for 30 min at 37 °C. Cells were fixed post-staining for 20 min with 4% formaldehyde at room temperature. Microscopic imaging (×60) was done with a Nikon A1 confocal microscope.

## Drug treatment

Both mouse neurons and patient iPSC-derived neurons were treated prior to harvest with 300 nM BafilomycinA1 (BafA1)[90] (Sigma-Aldrich, SML1661) or 300 nM Rapamycin[84] (LC Laboratories, Cat# R-5000) was administered to mouse or human neurons for 6 h or 24 h, respectively, prior fixation for morphology analysis or collection for protein analysis. Human neurons were treated with 100 nM Rapamycin for 1 week for electrophysiology assays.

## Generation of idCas9A-H9 hESC line

Human embryonic stem cell line H9 (WAe009-A)[36] was obtained from WiCell (Madison, WI). Culture of hESCs and genome editing were carried out using published methods[31,33]. Briefly, H9 hESCs were dissociated to single cells with TrypLE express (Thermo Fisher Scientific) and washed. $2–4 \times 10^6$ cells were electroporated (Gene Pulser Xcell, Bio-Rad; 250 V, 500 μF, 4 mm cuvette, infinite resistance) using 15 μg of plasmid 49535-*sgAAVS1*-T2 and 15 μg of donor plasmid AAVS1-TRE3G-dCas9-10xGCN (see plasmids section for plasmid

construction). Because the Cas9-sgRNA plasmid carries a puromycin-resistant gene, cells were transiently selected with puromycin (Thermo Fisher Scientific; 0.5 μg/ml 48–72 h after electroporation and 0.25 μg/ml 72–96 h afterward). After 2 weeks of selection, colonies were picked for expansion and screened by PCR for dCas9 integration. Positive (transgene-containing) clones were further validated by PCR and sequencing for the presence of correct gene targeting and five clones were identified to have correct gene targeting. The selected idCas9A-H9 #6 clone was further evaluated for karyotype and positive pluripotency markers.

## Generation and culturing of human pluripotent stem cells from 5q13.2trip ASD patient

The patient (WC-057-ASD) iPSC line was generated from the skin fibroblasts derived from the 5q13.2trip patient using a published method[31]. Briefly, the reprogramming of fibroblasts was done using the Cytotune 2.0 kit (ThermoScientific, Catalog #A16517) following the manufacturer's protocol. Cells were cultured on irradiated mouse embryonic fibroblasts (MEF) feeder layers (WiCell) with a daily change of hESC medium of DMEM/F12 (Thermo Fisher Scientific), 20% knockout serum replacement (KSR, Thermo Fisher Scientific), 0.1 mM 2-mercaptoethanol (Sigma-Aldrich), 1x L-Glutamine (Thermo Fisher Scientific), 6 ng/ml FGF-2 (Waisman Biomanufacturing Center). Individual reprogrammed colonies were picked and plated onto Matrigel for expansion and banking. iPSCs were stored in liquid $N_2$. Before starting differentiation, cells were thawed onto Matrigel-coated plates in mTeSR Plus (Stemcell Technologies) and cells were then lifted using 0.5 mM EDTA to transfer them back to MEFs. iPSCs grown on MEF were passaged by treatment with either dispase (Thermo Fisher Scientific) or collagenase (Thermo Fisher Scientific) in hESC medium, washed, and replated at a dilution of 1:5 to 1:10. G-banding was performed by WiCell Cytogenetics Services (Madison, WI), using published methods[31,33].

## Human PSCs and neural differentiation

Neural differentiation of human PSCs was carried out using either a dual SMAD inhibition method[31,33] or induce differentiation using published methods[38]. The control human GM1 iPSCs[31], WC5907 and WC6007 iPSCs[35], FXS patient-derived FXS1 (FX11-7) iPSCs[34] and FXS2 (FX13-2) iPSCs[34], and ASD patient line were differentiated using both methods. The WC5907 iPSCs[35], H9 ESCs[36], and H1 ESCs[36] were differentiated using the dual SMAD inhibition method. The idCas9A-H9 hESC line was differentiated using the NGN2-induced differentiation method. A summary of all hPSC lines and their shortened names used in the text are provided in Supplementary Data 1.

For the dual SMAD inhibition method, we followed published protocols[31,33]. Briefly, hPSCs were plated onto MEFs for 5 days, and neural differentiation was then induced by switching the hESC medium to neural induction medium (NIM) [DMEM/F12: Neurobasal 1:1, supplemented with 1× N2, 1× L-Glutamine, 1× Anti-Anti (GIBCO), 10 μM SB432542 (Selleck), 100 nM LDN193189 (Selleck), and 2 μM XAV-939 (Selleck)]. Cells were cultured in NIM for 9 days with a daily medium change. Cells were then dissociated with TrypLE and re-plated 1:1 on Matrigel-coated plates in neural progenitor cell (NPC) medium [Neurobasal medium, 1x L-Glutamine (Thermo Fisher Scientific), 1× N2, 0.5x B27 without vitamin A (Thermo Fisher Scientific), 1x Anti-Anti) supplemented with 10 μM ROCK inhibitor (Y-27632 dihydrochloride, Tocris)]. Cells were patterned for 7 days with daily change of NPC medium. The NPCs were re-plated for the differentiation of neurons. For neuronal differentiation, NPCs were dissociated with Accutase (Thermo Fisher Scientific) and re-plated on Matrigel-coated plates in NPC medium supplemented with 10 μM ROCK inhibitor and 0.1 μM Compound E (Calbiochem). After 24 hours, medium was changed to NDM medium [(NPC medium plus 200 μM ascorbic acid (Sigma-Aldrich), 1 μM cAMP (Sigma-Aldrich), 10 ng/ml BDNF (Peprotech), and

10 ng/ml GDNF (Peprotech) and 30% sucrose, supplemented with 0.1 μM Compound E (Calbiochem)]. NDM medium was changed twice every week.

For NGN2-induced differentiation, we followed published protocol[38]. Briefly, hPSCs were differentiated into NPCs using similar methods for the 5q13.2trip ASD iPSC line. The NPCs were then co-infected with TetO-NGN2-Neo and reverse tetracycline-controlled transactivator (rtTA) with modification. Briefly, NPCs were plated at a density of 250,000 cells/cm$^2$ in NPCs medium (DMEM/F12, 1× N2, 1× B27 without vitamin A (Thermo Fisher Scientific), 1× Anti-Anti, 20 ng/ml FGF2, 1 μg/ml mouse laminin (Thermo Fisher Scientific) supplemented with rock inhibitor (Y-27632 dihydrochloride, Tocris). To transduce NPCs, the media was aspirated and replaced with NPCs medium containing the lentiviruses LV-TetO-NGN2-Neo and LV-rtTA the following day. Viral media was replaced with fresh NPCs medium containing 2 μg/ml doxycycline (Thermo Fisher Scientific) after 24 h. On the following day, G418 selection media was replaced with fresh NPCs medium containing 400 μg/ml G418 (Sigma-Aldrich) and 1 μg/ml doxycycline for 3 days. After G418 selection, cells were dissociated with Accutase (Thermo Fisher Scientific) and re-plated on Matrigel-coated plates in Neurons medium (DMEM/F12, 1x N2, 1x B27 without vitamin A, 1x Anti-Anti, 20 ng/ml BDNF (Peprotech), 20 ng/ml GDNF (Peprotech), 500 ng/ml cAMP (Sigma-Aldrich), 200 μM ascorbic acid (Sigma-Aldrich), 1 μg/ml mouse Laminin (Thermo Fisher Scientific). Neurons were collected after 1 week for Western Blot, ICC, and morphology analysis. Electrophysiology analysis was done between 2 weeks and 6 weeks of differentiation.

### Isolation of mouse astrocytes
Mouse astrocytes were isolated from the telencephalon of newborn wild-type C57BL/6J mice using a published protocol[91]. Briefly, P0-P3 mouse forebrain was dissected and digested with trypsin for 5 min, cells were then dissociated by harsh trituration to avoid growing of neurons and plated onto T25 flasks in DMEM supplemented with 10% FBS. The isolated cells were plated in T25 flasks and cultured in a 37 °C, 5% CO$_2$ incubator until cells reached confluence. The flasks were then shaken at 100 rotations/min for 2 days at 37 °C to shake off proliferating cells and neurons. The media were changed. The cells were passaged a total of three times before the astrocytes were used for coculture experiments with human neurons.

### Multiwell microelectrode array (MEA) analysis
MEA analysis for human neurons was carried out using published methods[92] with modifications. Briefly, human NPCs (200,000 cells) were plated in 96-well MEA plates coated with poly-D-lysine and Matrigel for neuronal differentiation as described above. The idCas9A-H9 ESC line, 5q13.2trp ASD patient iPSC line, GM1 iPSC line, WC6007 iPSC line, and FXS patient iPSC lines (FXS1 and FXS2) were differentiated using NGN2-directed differentiation method for MEA. Then, 3 days after replating, LV-sgMAP1B or LV-sgCtrl was added to idCas9A-H9 neurons and LV-shMAP1B or LV-shNC was added to ASD patient iPSC-derived and FXS patient iPSC-derived neurons. One week after replating into MEA plates, mouse astrocytes were added onto cultured human Neurons medium supplemented with 10% FBS. After 48 h of co-culture, 50% of the medium in each well was exchanged, followed by twice a week media change. MEA recording of human neurons was performed weekly between 2 weeks and 6 weeks after differentiation. Neurons start showing activities at 2 weeks of differentiation. Since PSC lines differentiated exhibit activities at different time points, the 6-week data were shown for idCas9A-H9 neurons, and the 3-week data were shown for both ASD patient neurons and FXS neurons. Extracellular recordings were performed in a Maestro MEA system equipped with AxIS software (Axion Biosystems) using a bandwidth with a filter for 200 Hz to 3 kHz cutoff frequencies. Spike detection was performed using an adaptive threshold set to 6 times the standard deviation of the estimated noise on each electrode. Each plate was acclimatized for 5 min in the Maestro Instrument and recorded for 10 min for quantification. Recordings were performed before media change. MEA analysis was performed using the Axion Biosystems Neural Metrics Tool and Metric Plotting Tool. An electrode was considered active at a threshold of 5 spikes/min. The sample size for each treatment was indicated in figure legends. Wells with <25% active electrodes or MFR < 0.1 Hz were not included in the data analysis. Furthermore, wells that displayed insufficient quality, for example, cell clumping, were discarded based on literatures[93,94].

### Whole-cell recording of human neurons
Whole-cell patch-clamp recordings of human neurons were performed as described[95]. The idCas9A-H9 neurons were recorded starting from 4 weeks of differentiation. The ASD patient neurons were recorded starting from 2 weeks of differentiation. Briefly, the recording was made using a Multipatch 700B preamplifier, digitized with a Digidata 1440 A, and acquired using pClamp 10 (Molecular Devices). The extracellular solution contained (in mM): 119 NaCl, 5 KCl, 1.3 MgCl$_2$, 2 CaCl$_2$, 20 glucose, and 20 HEPES at pH = 7.4. In some experiments, the bath solution also contained 1 μM TTX, 20 μM bicuculline, and 50 μM D-AP5 to record the AMPA receptor-mediated miniature excitatory postsynaptic currents (mEPSCs) only. For mEPSC recordings, pipettes were filled with (in mM): 115 Cs-methanesulfonate, 25 TEA-Cl, 10 HEPES, 10 QX-314, 4 NaCl, 4 Mg-ATP, 1 EGTA, 0.3 Li-GTP, and 10 phospho-creatine, at pH = 7.2. For voltage-clamp experiments, all neurons were held at −70 mV. Series resistance was typically within 10−30 MΩ. mEPSC events were analyzed using Python. Events of magnitude less than three times the standard deviation (std x 3) of baseline or less than 5 pA were rejected. Artifacts were screened through visual inspection and overlapping events (event frequency of ≥100 Hz) were discarded. Raw current traces of 300 s were analyzed for peak amplitude and frequency which were averaged for each cell and used for further statistical analysis. To evaluate the neuronal intrinsic excitability, currents were injected into neurons in current clamp mode, and the frequency of evoked action potentials was quantified. Rheobase was measured by applying 1.5 s steps in increments of 5 pA until an action potential was generated. The pipette solution for this experiment contained the following (in mM): 140 K-gluconate, 7.5 KCl, 10 HEPES-K, 0.5 EGTA-K, 4 Mg-ATP, and 0.3 Li-GTP.

### Quantitative analysis of IHC/ICC signal intensity
The signal intensity of MAP1B in CAMK2A+ cells in the PFC of each animal was quantified using FIJI (Image J) software using published methods[39,65]. The z-stack images (2 μm interval) were acquired using Nikon A1 confocal microscope. The fluorescent intensity of MAP1B was measured after subtracting background pixel intensity in the same image using FIJI (Image J) software. Samples from four individual animals, each from a different litter were analyzed for each experimental condition.

The signal intensity of LC3 in primary neurons was quantified using FIJI (Image J) software using published methods[39,65] as stated above. All images for human neurons were acquired using Nikon A1 confocal microscope. Two batches of the mouse primary neuron images were acquired using a Zeiss Apotome microscope and another batch was acquired using Nikon A1 confocal microscope, the intensity in MAP1B-EE (LV-sgMAP1B infected) group was normalized to the Ctrl group (LV-sgCtrl infected) before doing statistical analysis.

### Puncta quantification
Images of p62 puncta, PLA puncta, mCherry-GFP-LC3 puncta, and autophagic vacuoles were acquired using Nikon A1 confocal microscope with ×60 oil objective. 3D reconstruction was conducted using FIJI (Image J) software with a Clear Volume plug-in. The numbers of puncta in each cell were counted manually by

experimenters who were blind to the identity and treatments of the samples.

## Statistical analysis

The sample size was determined based on power analyses (StateMate), our publications[31,33,39,65,82,96], and literature[97–101]. Our sample sizes are similar to those reported in previous publications (see citations within each procedure). Data distribution was assumed to be normal but this was not formally tested. Statistical analysis was performed using ANOVA and Student's *t*-test, unless specified, with the aid of SPSS version 28 and GraphPad Prism 9. An unpaired two-tailed Student *t*-test was used for two-group comparisons. One-way ANOVA was used for the analysis of multiple experimental treatments compared to one control condition. Two-way ANOVA with Bonferroni post hoc test was used for analyzing multiple groups. Sholl analysis was carried out using multivariate analysis of variance (MANOVA) using SPSS statistical software. Outliers were identified using GraphPad Prism 9 software with the ROUT method (Q = 1%). Probabilities of $p < 0.05$ were considered significant. Please see Supplementary Data 1 for the cell lines, animal models, and tissue samples used in this study as well as the replicates for different models.

## Reporting summary

Further information on research design is available in the Nature Portfolio Reporting Summary linked to this article.

## Data availability

The data generated in this study are provided in the Supplementary Information/Source Data file submitted with this paper. Source data are provided with this paper.

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

## Acknowledgements

We thank Y. Xing, M. Keefe, and T. Korabelnikov in the Zhao lab for technical assistance; J. Panksepp, M. Eastwood, D. Bolling, and K. Knobel at the Waisman IDD Model Core for core services; Y. Yin, C. Soref, Dr. A. Bhattacharyya, Dr. G. Rice, and Waisman Clinical Translational Core for assistance in the generation of patient-derived iPSCs; Dr. Heather Simmons, Director of Pathology Services, WNPRC, for non-human primate tissue acquisition. This work was supported by grants from the National Institutes of Health (R01MH118827, R01NS105200, and R01MH116582 to X.Z., R01HD064743 to Q.C., R01NS064025, R01AG067025, and U01MH116492 to D.W., P51 OD011106 to the Wisconsin National Primate Research Center, U54HD090256 and P50HD105353 to the Waisman Center, R24HD000836 to I.A.G.), DOD IIRA W81XWH2210621 grant, Brain Research Foundation, UW Vilas (Mid-Career Award), Wisconsin Alumni Research Foundation, Jenni and Kyle Professorship (to X.Z.), National Science Foundation Career Award 2144475 to D.W., NARSAD Young Investigator Grant from the Brain and Behavior Research Foundation 28721 (to A.M.M.S.), T32 GM141013 Molecular Pharmacology training grant and SciMed scholarship (to N.M.M.-A.), Simons SFARI pilot grant (to X.Z., A.M.M.S., Q.C., and D.W.), FRAXA Postdoctoral fellowships (to M.S. and C.L.S.), postdoctoral fellowships from UW Stem Cell and Regenerative Medicine Center and Autism Science Foundation (to C.L.S.). Hilldale Undergraduate Research Scholarships (to S.X.H. and E.D.J.), R24HD000836 (to Birth Defects Research Laboratory).

## Author contributions

X.Z. conceived the concept and designed experiments. Y.Guo. designed and performed experiments, collected data, and analyzed data. Y.Guo. and X.Z. wrote the manuscript. M.S. performed behavior tests and cortical slice experiments. M.S., A.M.M.S., and Y.Gao generated cortical slice data. N.M.M.-A., J.L., M.L., S.X.H., E.D.J., K.A.S., M.E.S., and C.L.S. collected data. Q.D. and Q.C. performed electrophysiological analyses. V.L.H. performed ASD patient microarray analysis. D.W. performed gene expression analysis using PsychENCODE data; Birth Defects Research Laboratory contributed to human cortical slice analysis. J.E.L. contributed to rhesus macaque analysis.

## Competing interests

The authors declare no competing interests.

## Additional information

## Birth Defects Research Laboratory

**Ian A. Glass[9] & Dan Doherty[9]**

[9]Birth Defects Research Laboratory, University of Washington and Seattle Children's Research Hospital, Seattle, WA 98195, USA. A full list of members and their affiliations appears in the Supplementary Information.

