## [Peer Review File · Nature Communications]

Elevated levels of FMRP-target MAP1B impair human and mouse neuronal development and mouse social behaviors via autophagy pathwayEditorial Note: Parts of this Peer Review File have been redacted as indicated to remove third-party material where no permission to publish could be obtained.

Reviewers' Comments:

Reviewer #1:

Remarks to the Author:

Guo et al. "Elevated levels of FMRP-target MAP1B impair neuronal development and social behaviors via autophagy pathway"

Microtubule associated protein MAP1B mRNA is a target of the Fragile X messenger ribonucleoprotein 1 protein. Previous studies have suggested that MAP1B is causally linked to the pathogenesis of Fragile X syndrome - but as the authors point out, such studies have mainly investigated the impact of MAP1B levels through the expression of very high levels of MAP1B.

This manuscript investigates the impact of mildly elevated levels of the FMRP-target MAP1B on neuronal development using human-, macaque- and mouse-derived model systems.

The authors demonstrate that shRNA mediated knockdown of FMRP1 in human and macaque cortical slice cultures results in mildly elevated MAP1B. Similarly iPSC derived neurons from FXS show higher levels of MAP1B, hence confirming previous studies indicating dysregulation of MAP1B in FXS / regulation of MAP1B levels by FMRP1. The authors use a dCas9-Activator strategy to achieve 2-3 fold increased expression of MAP1B in human iPSC -derived neurons and in the murine prefrontal cortex (PFC). Using this strategy they find that increased MAP1B levels impair neurite growth and alter neurophysiological properties (increased firing rate, higher burst rate, higher excitability). Moreover, they demonstrate that in mice with a mild increase in MAP1B in the PFC, social behavior is impaired. In a parallel set of experiments the authors analyze iPSCs from a subject with autism, who carries a small 467.5-kilobase (kb) triplication of chromosome 5q13.2 containing the MAP1B gene.

Interestingly, reducing MAP1B levels in the respective iPSC neurons alters neuronal development and electrophysiological parameters in a direction (increased dendrite complexity, decreased firing rate, higher burst rate, higher excitability) that together with the dCas9 activator data indicates a pathophysiological involvement of MAP1B levels. In search of the mechanism the authors focus on the previously described role of MAP1B in modulating autophagy-lysosomal pathway activity. Using the mouse dCas9 system they confirm that increased MAP1B levels inhibit autophagy-lysosomal pathway activity, most likely by inhibition of autophagosome formation. Analyses of the LC3 levels in human and macaque neurons with increased MAP1B expression, FMRP1 knockdown, or triplication of chromosome 5q13.2 support the idea of MAP1B inhibition of autophagy-lysosomal pathway activity. The authors also provide evidence for the previously described phenomenon that MAP1B sequesters LC3I thereby reducing autophagosome formation. Finally, the authors demonstrate that rapamycin treatment ameliorates neuronal development and neurophysiological deficits in FXS and triplication of chromosome 5q13.2 iPSC-derived neurons.

MAP1B has been previously implied in FXS pathophysiology as well as in the modulation of autophagy; autophagy on the other hand has been linked to the pathophysiology of autism spectrum disorders and FXS. In that regard the manuscript does not necessarily offer a very major conceptual advancement. Nevertheless, I consider this study very important for the field because it convincingly connects these dots and elegantly demonstrates for the first time that mildly elevated levels of MAP1B do generate a neurodevelopmental phenotype. The study is very well conceived; the combination of model systems not only exploits their distinct strengths but also indicates that MAP1B levels and their impact on neuronal development is a generalized mechanism across species.

I have one few major concerns.

The analysis of autophagy deficits in human cellular/slice culture model systems (Fig.5) is quite superficial and solely based on the number of LC3 intensities. These should be complemented with p62 and (in the case of the iPSC derived neurons) with some biochemical or reporter assays for autophagy-lysosomal activity.

For the rescue using rapamycin and MAP1B knockdown, confirmation of restored autophagy-lysosomal activity should be included.

Reviewer #2:

Remarks to the Author:

In the manuscript submitted by Guo and colleagues, the authors aim to show the functional role of microtubule-associated protein 1B (MAP1B) on the development of neurons in 2 forms of neurodevelopmental delay – Fragile X Syndrome and 5q13.2 triplication syndrome. To achieve this the authors examine the function of MAP1B in neurons derived from foetal brain tissue, iPSCs and rodents to show that excessive MAP1B observed in these two models of ASD can lead to impaired dendritic structure and altered cellular and synaptic function. The panoply of experiments used are cutting-edge and would be suitable to address this question and overall the evidence they present is generally indicative that their conclusions are reasonable. The manuscript is extremely well written and easy to follow. I have one major issue that is that the majority of the data from FXS and ASD cells have been collected from 1 FXS and 1 ASD individual. This makes generalisation to the human population at large - tenuous at best. All data should be presented in the context of how many biological replicates the data was actually derived from.

Major:

1) My major concern with this study is the author's selection of appropriate replicates and their statistical comparisons and tests performed. Whilst I appreciate that individual culture preparations/iPSC passages are time-consuming, it is not a fair comparison to present the values produced from single cells as the chosen replicate for any of the data that the authors present. Many of the effect sizes that their data display are relatively small, as such the use of pseudoreplicates (and not the biological replicate) for statistical comparison dramatically increase the likelihood of type-1 statistical errors (inappropriate rejection of the null-hypothesis). As such, whilst the authors clearly have performed a technically challenging and interesting set of experiments, these results have limited biological scope.

As a minimum the authors must report the average responses of individual iPSC lines as a single biological replicate for each experiment. It is not be a fair comparison to then perform statistical analysis on these single replicates as the authors have performed. Where the authors do show multiple biological replicates (control iPSCs, human slice cultures, and rodent work), the data should only be statistically analysed from the level of the biological individual. This is most pertinent for the iPSC work, where 5q13.2trip and FXS neurons were only derived from 1 individual each. This risks skewing the data towards the effects observed in an individual and inform little of how relative their observations may be to the FXS or ASD communities at large. The appropriate level of comparison has been performed elsewhere (Das Sharma et al., 2020 Molecular Autism), so such additional cell lines do exist and could be used. As far as I can see your data for FXS and ASD lines is essentially N=1 and should be treated as such. Showing within line variability is OK – but you need to clearly state that this data relates to neurons derived from a single individual, then avoid sweeping generalisations about the role of MAP1B in FXS or ASD generally. For groups with more than 3 replicates, the authors could generate a more complex statistical approach incorporating the use of linear mixed-effects models (or their generalised form) to account for intra-subject variability. Such approaches will reduce the perceived power/statistical differences from their reported data and may require the collection of further data from multiple individuals.

Beyond this, the actual mechanics of how the statistical analyses were performed leave a little to be desired. As stated in the methods: "No statistical methods were used to pre-determine sample sizes, but our sample sizes are similar to those reported in previous publications (see citations within each procedure). Data distribution was assumed to be normal but this was not formally tested". For most data sets the sample size is 1-4 biological replicates – arguably below levels that should be statistically tested. Even overlooking this fact and the repeated use of inappropriate replicate, most of the data sets shown look to be non-Gaussian or even bi-modal (worrying when only 2 biological replicates have been used). These must be tested with the appropriate replicate and statistic used.

2) There is no evidence that FMRP is actually reduced in your human ex vivo cortical slices (Figure 1). As a minimum quantify protein expression by western blot or FMRP immunolabelling .

3) The western blot analyses presented in Figure 2D, 4D, and 6B-D is unconvincing. First, given the previous indication of the inappropriate use of statistical analysis, the Western blot (when quantified) should be subject to either 2-way ANOVA with genotype/protein used as variables – and multiple-comparisons accounted for in post-hoc tests (if interaction effect observed).

I suspect that the use of repeated paired tests has artificially boosted the reported statistical differences – which likely over-estimate biological effects.

Second, for Figure 2D there is no quantification of the relative increase in MAP1B levels – which makes interpretation difficult from a single gel/blot image.

4) For Current-Frequency plot analysis (as shown in Figures 1L, 2L, 7J, the use of 2-WAY RM ANOVA for such comparisons is not appropriate, as this is prone to a) pseudoreplication, and b) assumes that increasing current steps are true repeated measures (which they aren't). A more fair comparison would be to measure the rheobase current threshold and the slope of the relationship in cells. These individual metrics allow for more fair comparison of the firing curves of neurons.

5) For the rapamycin treatment experiments, the statement “Indeed, rapamycin treatment of ASD patient neurons led to increased dendritic complexity and total dendritic length compared with control neurons (Fig. 7a-d)”. But, no direct comparison is made in these figures. A more comprehensive analysis comparing Control & ASD +/- rapamycin should be made with appropriate statistical comparisons performed. Such analysis is partially performed in supplementary figure 13. But this should be main comparison made and not just supplementary. Why was no such effect of rapamycin on control neurons made for physiological experiments?

6) I have a bit of a hard time interpreting the average total length of neurites from the shFMR1 is less than 50 micrometres (Figure 8H). Indeed, the maximum Sholl radius is 40 micrometres, and the example cells shown in Figure 8F all possess multiple neurites. This raises a few concerns:

a) Are these FMR1 neurons at the same maturation state as control or are they differentiated into neurons at this point? From the Rapamycin treatment group, clearly that many “neurons” also do not mature. Are these even neurons? What controls have been done to confirm that they have reached the same maturation state?

b) How do these fall out within different biological replicates (was this the sample with only 2 replicates?).

c) Could all this be a developmental issue (see below) and perhaps assessing dendrite length a few days/week later might have allowed development to have occurred a bit more.

d) From the example reconstruction images shown the total dendritic length extent appears greater than this. Including some wide-field images from those shown in Figure 8F where multiple cells can be seen (without overlain tracings) would allow for a fair comparison.

7) From the methods and results is not abundantly clear when the iSPCs or slices were routinely taken for imaging. There is evidence (especially in FXS) that neurite outgrowth is slower than control neurons (Doers et al., 2014) – if you tested say a week later – would these deficits be normalised out and would the ASD models have caught up with control? Including 1 time point later in neuronal development would crucial to tie in with existing literature where dendritic structure in Fmr1-KO rodent models reaches typical lengths early in development (e.g. Bureau et al., 2008; Till et al., 2015).

8) As far as I can tell there is no direct comparison made between the dendritic morphology of

5q13.2trip neurons derived from iPSCs to control neurons – but rather the effect of individual genes found in this locus. If the logic is that this genetic mutation enhances MAP1B levels, one might expect that the dendritic lengths of these neurons differ to control. By eye, judging to the total neurite (dendritic) lengths between control neurons (Figure 1G) and 5q13.2trip neurons (Figure 2G) there is no difference in total length. This comparison is important and may impact the authors overall conclusion about the role of MAP1B enriched expression on neuronal structure. If 5q13.2trip neurons express endogenously higher levels of MAP1B it would be expected that their dendritic structure is more simplified than that of control neurons - based on the central hypothesis of this study. Simply looking at each of the genes in turn from this locus may overlook potential polygenic effects.

Minor:

- 1) Dendrite analysis: Did the authors confirm that the structures they reconstructed were actually dendrites – or are these neurites comprising both axonal and dendritic sections. This distinction should be stated.
- 2) Mini EPSC analysis: the phrase “Threshold was typically kept at –5 pA and adjusted with respect to the baseline noise.” Is quite nebulous. What was the average threshold used for each group and did it differ between tested groups? A minor arbitrary shift in threshold can wildly shift the number of events recorded, which may bias the data.
- 3) There has been much literature on the excitability of neurons in Fmr1 mice and rats during early postnatal development (e.g. Gibson et al., 2009, Pilpel et al. 2009, Luque et al., 2017 Domanski et al., 2019, Booker et al., 2019, Ordemann et al., 2021...), it might be useful to highlight some of these studies and causally link them to the physiological correlations you have made between synaptic function and MAP1B expression.
- 4) The introduction is very FXS focused and the 5q13.2trip approach is very minimally discussed. A more general background would be useful, such as the detail at the start of the second results section.

Point-by-Point Response to Reviewers for NCOMMS-22-41448

We thank the reviewers for their effort and constructive suggestions

REVIEWER COMMENTS

Reviewer #1:

Guo et al. "Elevated levels of FMRP-target MAP1B impair neuronal development and social behaviors via autophagy pathway"

Microtubule associated protein MAP1B mRNA is a target of the Fragile X messenger ribonucleoprotein 1 protein. Previous studies have suggested that MAP1B is causally linked to the pathogenesis of Fragile X syndrome - but as the authors point out, such studies have mainly investigated the impact of MAP1B levels through the expression of very high levels of MAP1B.

This manuscript investigates the impact of mildly elevated levels of the FMRP-target MAP1B on neuronal development using human-, macaque- and mouse-derived model systems.

The authors demonstrate that shRNA mediated knockdown of FMRP1 in human and macaque cortical slice cultures results in mildly elevated MAP1B. Similarly iPSC derived neurons from FXS show higher levels of MAP1B, hence confirming previous studies indicating dysregulation of MAP1B in FXS / regulation of MAP1B levels by FMRP1. The authors use a dCas9-Activator strategy to achieve 2-3 fold increased expression of MAP1B in human iPSC -derived neurons and in the murine prefrontal cortex (PFC). Using this strategy they find that increased MAP1B levels impair neurite growth and alter neurophysiological properties (increased firing rate, higher burst rate, higher excitability). Moreover, they demonstrate that in mice with a mild increase in MAP1B in the PFC, social behavior is impaired. In a parallel set of experiments the authors analyze iPSCs from a subject with autism, who carries a small 467.5-kilobase (kb) triplication of chromosome 5q13.2 containing the MAP1B gene. Interestingly, reducing MAP1B levels in the respective iPSC neurons alters neuronal development and electrophysiological parameters in a direction (increased dendrite complexity, decreased firing rate, higher burst rate, higher excitability) that together with the dCas9 activator data indicates a pathophysiological involvement of MAP1B levels. In search of the mechanism the authors focus on the previously described role of MAP1B in modulating autophagy-lysosomal pathway activity. Using the mouse dCas9 system they confirm that increased MAP1B levels inhibit autophagy-lysosomal pathway activity, most likely by inhibition of autophagosome formation. Analyses of the LC3 levels in human and macaque neurons with increased MAP1B expression, FMRP1 knockdown, or triplication of chromosome 5q13.2 support the idea of MAP1B inhibition of autophagy-lysosomal pathway activity. The authors also provide evidence for the previously described phenomenon that MAP1B sequesters LC3I thereby reducing autophagosome formation. Finally, the authors demonstrate that rapamycin treatment ameliorates neuronal development and neurophysiological deficits in FXS and triplication of chromosome 5q13.2 iPSC-derived neurons.

MAP1B has been previously implied in FXS pathophysiology as well as in the modulation of autophagy; autophagy on the other hand has been linked to the pathophysiology of autism spectrum disorders and FXS. In that regard the manuscript does not necessarily offer a very major conceptual advancement. Nevertheless, I consider this study very important for the field because it convincingly connects these dots and elegantly demonstrates for the first time that mildly elevated levels of MAP1B do generate a neurodevelopmental phenotype. The study is very well conceived; the combination of model systems not only exploits their distinct strengths but also indicates that MAP1B levels and their impact on neuronal development is a generalized mechanism across species.

Response: we thank this reviewer for positive evaluation of our manuscript. We have made significant improvement as detailed below. The text with significant changes are blue. The new data are included in the **revised panels (red labels)** of figures. The data with revised statistical analysis are labeled **blue** (blue labels) in figures.

I have one few major concerns.

The analysis of autophagy deficits in human cellular/slice culture model systems (Fig.5) is quite superficial and solely based on the number of LC3 intensities. These should be complemented with p62 and (in the case of the iPSC derived neurons) with some biochemical or reporter assays for autophagy-lysosomal activity.

For the rescue using rapamycin and MAP1B knockdown, confirmation of restored autophagy-lysosomal activity should be included.

Response:

To address the reviewer's concerns, we have performed p62 puncta counting in idCas9A-H9 neurons (Ctrl, MAP1B-EE, Fig. S9), ASD neurons (Ctrl, MAP1B knockdown or RAP treatment, Fig. S11a, b and Fig. S16d, e), human ex vivo cortical slices (Fig. S18a, b).

Because the reviewer suggested that we confirm restored autophagy-lysosomal activities by rapamycin and MAP1B knockdown, we added a fluorescence-based autophagy vacuole assay (Abcam, ab139484) to assess autophagy activities in the rapamycin and MAP1B knockdown rescue experiments in ASD neurons (Fig. S11c, d and Fig S16f, g), and in FXS neurons (FXS1, FXS2: Ctrl, MAP1B knockdown or RAP treatment, Fig. S17i-l). Because our idCas9A-H9 neurons express GFP and the autophagy detection dye is green, we could not perform this assay for idCas9A-H9 neurons. Our new results have confirmed that Elevated MAP1B in either ASD

neurons or FXS neurons leads to autophagy deficits and rapamycin treatment and MAP1B knockdown restore autophagy activity in human neurons.

Reviewer #2:

In the manuscript submitted by Guo and colleagues, the authors aim to show the functional role of microtubule-associated protein 1B (MAP1B) on the development of neurons in 2 forms of neurodevelopmental delay – Fragile X Syndrome and 5q13.2 triplication syndrome. To achieve this the authors examine the function of MAP1B in neurons derived from foetal brain tissue, iPSCs and rodents to show that excessive MAP1B observed in these two models of ASD can lead to impaired dendritic structure and altered cellular and synaptic function. The panoply of experiments used are cutting-edge and would be suitable to address this question and overall the evidence they present is generally indicative that their conclusions are reasonable. The manuscript is extremely well written and easy to follow. I have one major issue that is that the majority of the data from FXS and ASD cells have been collected from 1 FXS and 1 ASD individual. This makes generalisation to the human population at large - tenuous at best. All data should be presented in the context of how many biological replicates the data was actually derived from.

Response: we thank this reviewer for positive evaluation of our manuscript. We also thank this reviewer for constructive suggestions to help strengthen our study.

Major:

1) My major concern with this study is the author's selection of appropriate replicates and their

statistical comparisons and tests performed. Whilst I appreciate that individual culture preparations/iPSC passages are time-consuming, it is not a fair comparison to present the values produced from single cells as the chosen replicate for any of the data that the authors present. Many of the effect sizes that their data display are relatively small, as such the use of pseudoreplicates (and not the biological replicate) for statistical comparison dramatically increase the likelihood of type-1 statistical errors (inappropriate rejection of the null-hypothesis). As such, whilst the authors clearly have performed a technically challenging and interesting set of experiments, these results have limited biological scope.

As a minimum the authors must report the average responses of individual iPSC lines as a single biological replicate for each experiment. It is not a fair comparison to then perform statistical analysis on these single replicates as the authors have performed. Where the authors do show multiple biological replicates (control iPSCs, human slice cultures, and rodent work), the data should only be statistically analysed from the level of the biological individual. This is most pertinent for the iPSC work, where 5q13.2trip and FXS neurons were only derived from 1 individual each. This risks skewing the data towards the effects observed in an individual and inform little of how relative their observations may be to the FXS or ASD communities at large. The appropriate level of comparison has been performed elsewhere (Das Sharma et al., 2020 Molecular Autism), so such additional cell lines do exist and could be used. As far as I can see your data for FXS and ASD lines is essentially N=1 and should be treated as such. Showing within line variability is OK – but you need to clearly state that this data relates to neurons derived from a single individual, then avoid sweeping generalisations about the role of MAP1B in FXS or ASD generally. For groups with more than 3 replicates, the authors could generate a more complex statistical approach incorporating the use of linear mixed-effects models (or their generalised form) to account for intra-subject variability. Such approaches will reduce the perceived power/statistical differences from their reported data and may require the collection of further data from multiple individuals.

Beyond this, the actual mechanics of how the statistical analyses were performed leave a little to be desired. As stated in the methods: “No statistical methods were used to pre-determine sample sizes, but our sample sizes are similar to those reported in previous publications (see citations within each procedure). Data distribution was assumed to be normal but this was not formally tested”. For most data sets the sample size is 1-4 biological replicates – arguably below levels that should be statistically tested. Even overlooking this fact and the repeated use of inappropriate replicate, most of the data sets shown look to be non-Gaussian or even bi-modal (worrying when only 2 biological replicates have been used). These must be tested with the appropriate replicate and statistic used.

Response:

First, we agree with the reviewer about the need for biological replicates. We were only able to obtain cells from one ASD patient with 5q13.2trip. We indeed have identified several 5q13.2dup or trip individuals reported in DICEPHER and ClinVAR (see Supplementary Table 2). During the past several years, we have sent out requests to managers of those database and physicians who submitted the patient information. We have also reached out to Simons SFARI and our colleagues working with autism genetics (e.g. Stephen Scherer at Toronto SickKid). However, we could not obtain cells from any of those individuals to generate iPSCs. According to our colleagues, small size of copy number variants (like we observed in this ASD patients) were not reported in early studies due to low resolution of DNA microarrays, therefore the incidence of this CNV might be under-detected and under-reported in studies. We therefore created the

idCas9A-H9 line using human H9 ESCs, and used LV-sgNC infected idCas9A-H9 cells (no MAP1B-EE) as its isogenic control for LV-sgMAP1B infected idCas9A-H9 cells (with specific MAP1B-EE). We also used 5q13.2Trip ASD cells with MAP1B knockdown (*LV-shMAP1B* infection) as an isogenic control for 5q13.2Trip ASD cells without MAP1B knockdown (*LV-shNC* infection). Therefore in our paper, we used these two isogenic MAP1B-EE/control pairs to assess MAP1B functions in human neurons. In addition, our data from mouse neurons (>3 independent biological replicates) provide another set of “replicates” to support our conclusion. We hope that the reviewer agree that our data from two pairs of isogenic human neurons and >3 biologically independent mouse studies have provided sufficient support for our conclusion.

In addition, we have confirmed our key findings on neuronal dendritic complexity, autophagy activity, and neuronal excitability in another FXS iPSC line and control line (Please see Fig S1j-k, Fig. 8a-d, and Fig. S17). Therefore, the conclusion of our study is supported by data from both human FXS iPSCs (two lines with multiple differentiation each line) and developing human brain tissue (3 biological replicates) and macaque brain tissue (3 biological replicates) with FMRP knockdown.

Therefore, in our study, we used a combination of patient derived and gene edited hPSCs, human and macaque ex vivo brain slices, and mouse models to investigate the relationship between FMRP and MAP1B and how disease relevant elevations of MAP1B impact neuronal development and behaviors. We have revised the conclusion to emphasize the combination models we used to support our conclusion.

Second, about statistical analysis, we have carefully considered the reviewer's comments and read both the paper recommended by the reviewer and the latest publications in high impact journals especially Nature journals. Accordingly, we have revised our statistical analyses, following the principles described below:

I. We have changed our analysis and now use the mean value of each differentiation, primary neuron isolation or individual biological sample (rather than individual cell) as replicate for statistical analysis in most cases, with exceptions described in II, III, IV below. The figure panel with revised statistics have blue labels. Please see the detailed information on statistics in each figure legend.

II. For whole-cell patch clamp recording, we use data from individual neuron for statistical analysis, which is well-accepted in this field. Please see *Wang et al., 2022 Mol Psychiatry (PMID: 36280753)* and the paper recommended by the reviewer 2 *Das Sharma et al., 2020 Molecular Autism, (PMID: 32560741)*.

III. For Scholl analysis of neuronal morphology, single neuron data are used for multivariate ANOVA based on recent publications (*Li et al., Nature Neuroscience 2022, PMID:35524139; Nam et al., Nature communications 2022, PMID:35418126; Wang et al., Cell 2021; PMID:34758294*). In our revision, data from different batches are shown in different shapes.

Below: Dendritic morphology analysis in Fig 3n-p in Li et al., Nature Neuroscience 2022.

[Redacted]

m, Morphology of individual tdTomato⁺ cells at 16 dpi following optogenetic stimulation. Scale bar, 10 μ m. **n-p**, Sholl analysis (**n**), dendrite length (**o**) and total dendrite branches (**p**) of tdTomato⁺ cells at 16 dpi. $n = 54$ cells for YFP group and $n = 53$ cells for Chr2 group, $**P < 0.0001$ by two-sided unpaired t -test. Mol, molecular layer.

IV. For MEA recording, individual well was used to do analysis. MEA has multiple electrode in each well (8-64 depending on the format, we have 8/well for our 96-well plate format). Because neurons plated into each well have random chances to be recorded, each well is different from other wells. Most MEA studies define replicates as the number of independent wells, excluding wells with below threshold number of active electrode, according to a recent review article in this topic: *McCready et al., Biology2022, (PMID: 35205182: "Multielectrode Arrays for Functional Phenotyping of Neurons from Induced Pluripotent Stem Cell Models of Neurodevelopmental Disorders.")* This method of defining replicates is widely used by other researches (*Admas et al. Molecular psychiatry 2022; Mossink et al. Molecular psychiatry 2021, Que et al. The Journal of Neuroscience 2021*). In our revision, data from different batches of neurons are shown in different shapes of symbols to demonstrate that no one batch of neurons skews the data.

Below: MEA analysis in published Figure 5 d,f from *Admas et al. Molecular psychiatry 2022*.

[Redacted]

Furthermore, we considered the method used by the paper recommended by reviewer 2 (in Major point 1, *Das Sharma et al., Molecular Autism 2020*). Although both our FXS1 and FXS2 lines show hyperexcitability and responses to MAP1B knockdown and Rapamycin treatment, these two lines are different in levels of excitability and features. Therefore we believe it is more clear to provide analysis of these two lines in separate figures rather than combining them in the same figure. We do agree with the reviewer that we should clearly indicate the number of experimental replicates and biological replicates in each figure legend using n and N. We have also revised the Method section by stating “The number of experimental replicates is denoted as ‘n,’ while ‘N’ represents number of biological replicates from which ‘n’ is obtained.”

Finally, our statement “No statistical methods were used to pre-determine sample sizes, but our sample sizes are similar to those reported in previous publications (see citations within each procedure). Data distribution was assumed to be normal but this was not formally tested” was incorrect. We have revised this statement in the Methods to “The sample size were determined based on power analyses (StateMate), our publications , and literature”.

In addition, to clearly describe the replicates used in this study, we generate a supplementary table (Table S1) to describe the cell lines, animal models, and tissue samples we used in this study as well as the replicates for different models.

2) There is no evidence that FMRP is actually reduced in your human ex vivo cortical slices (Figure 1). As a minimum quantify protein expression by western blot or FMRP immunolabelling.

Response: Our shFMR1 was validated in transfected HEK293 cells. We agree with the reviewer that we should validate knockdown in ex vivo tissue. We did not provide these data because there was no good antibody for FMRP immunolabelling. We have recently found a good FMRP antibody, therefore we performed immunostaining for FMRP in the human ex vivo cortical slices. Please see Fig. S1c and d. The results showed that FMRP protein level was indeed reduced in LV-*shFMR1* infected brain slice.

3) The western blot analyses presented in Figure 2D, 4D, and 6B-D is unconvincing. First, given the previous indication of the inappropriate use of statistical analysis, the Western blot (when quantified) should be subject to **either 2-way ANOVA with genotype/protein used as variables – and multiple-comparisons accounted for in post-hoc tests (if interaction effect observed).**

I suspect that the use of repeated paired tests has artificially boosted the reported statistical differences – which likely over-estimate biological effects.

Second, for Figure 2D there is no quantification of the relative increase in MAP1B levels – which makes interpretation difficult from a single gel/blot image.

Response:

For Figure 2d, to address the reviewers' concerns, we added two more batches of differentiation for each hPSC line used for this figure and performed quantification, please see new Fig. 2d.

As for Fig. 4d, the original Fig. 4d was two experiments performed together:

One experiment was to determine whether reduced LC3-II levels in MAP1B EE is result of production of total LC3 protein levels or a result of decreased LC3-II vs LC3-I levels.

The second purpose is to determine whether reduced LC-3 in MAP1B-EE result from enhanced LC3 degradation through autophagosome-lysosome fusion, therefore we treated the neurons with BafA1 that specifically inhibits autophagosome-lysosome fusion, but not autophagosome formation.

These two experiments should not be analyzed together which causes confusion and misuse of statistical methods. We therefore performed more experiments separately and plotted/analyzed them separately please see new Fig. 4 d-g.

For Fig. 6b-d, we agree with the reviewer and have changed our analyses to use unpaired student *t* test, please see new Fig. 6c and d.

4) For Current-Frequency plot analysis (as shown in Figures 1L, 2L, 7J), the use of 2-WAY RM ANOVA for such comparisons is not appropriate, as this is prone to a) pseudoreplication, and b) assumes that increasing current steps are true repeated measures (which they aren't). A more fair comparison would be to measure the rheobase current threshold and the slope of the relationship in cells. These individual metrics allow for more fair comparison of the firing curves of neurons.

Response: We agree with the reviewer and measured the rheobase current threshold for Fig. 11, 2l, and 7j. Please see the new figures 11, 2l, and 7j.

5) For the rapamycin treatment experiments, the statement “Indeed, rapamycin treatment of ASD patient neurons led to increased dendritic complexity and total dendritic length compared with control neurons (Fig. 7a-d)”. But, no direct comparison is made in these figures. A more comprehensive analysis comparing Control & ASD +/- rapamycin should be made with appropriate statistical comparisons performed. Such analysis is partially performed in supplementary figure 13. But this should be main comparison made and not just supplementary. Why was no such effect of rapamycin on control neurons made for physiological experiments?

Response: The statement “Indeed, rapamycin treatment of ASD patient neurons led to increased dendritic complexity and total dendritic length compared with control neurons (Fig. 7a-d)” is not correctly stated. What we want to conclude here is “Indeed, rapamycin treatment of ASD patient neurons led to increased dendritic complexity and total dendritic length compared with vehicle treated ASD neurons”. We have changed the statement in the main text.

To address the reviewer’s concerns, we performed a direct comparison between Control neurons and ASD neurons with/without rapamycin treatment. Please see Fig. S16 a-c. The results indicate that the ASD neurons with elevated levels of MAP1B have reduced dendritic complexity than control neurons, supporting that elevated MAP1B expression led to neuronal morphological deficits. Since the main figures are already quite full, we included these data in the supplementary figures.

In addition, before we treated neurons with rapamycin, we searched literatures (*Martin et al., 2020 Molecular Autism, PMID: 31921404; Crutcher et al., 2019 Sci Rep, PMID: 31685878*) and selected a concentration of rapamycin (300mM for 24h treatment for tracing, 100mM for long-term treatment) that has a minimal effect on healthy (control) human neurons but has significant effect on human neurons of disease models. For example, in *Martin et al., 2020 Molecular Autism* (see their fig. 2f), the authors did not observe significant effect of rapamycin treatment had an effect on neurite number or length in their control human neurons.

- 6) I have a bit of a hard time interpreting the average total length of neurites from the shFMR1 is less than 50 micrometres (**Figure 8H**). Indeed, **the maximum Sholl radius is 40 micrometres**, and the example cells shown in Figure 8F all possess multiple neurites. This raises a few concerns:
- Are these FMR1 neurons at the same maturation state as control or are they differentiated into neurons at this point? From the Rapamycin treatment group, clearly that many “neurons” also do not mature. Are these even neurons? What controls have been done to confirm that they have reached the same maturation state?
 - How do these fall out within different biological replicates (was this the sample with only 2 replicates?).
 - Could all this be a developmental issue (see below) and perhaps assessing dendrite length a few days/week later might have allowed development to have occurred a bit more.
 - From the example reconstruction images shown the total dendritic length extent appears greater than this. Including some wide-field images from those shown in Figure 8F where multiple cells can be seen (without overlain tracings) would allow for a fair comparison.

Response:

We address the points a)-d) below:

a). To address the concerns of this reviewer, we have performed staining in human ex vivo brain slice using antibodies for two neuronal markers, MAP2 and NeuN staining (**Fig. S1a, b**). Our quantitative results showed that among lentivirus infected cells, 87% are MAP2+ and 92% are NeuN+. Only the cells with at least one process were chosen for tracing. Most neurons are quite immature in midfetal cortex, which is similar to hPSC differentiated neurons. We have included these information in the revised text and Methods.

b). These results were obtained from 3 independent biological replicates (cortices from 3 different postmortem brains). See new Table S1.

c). The oldest human tissue without any known genetic deficits that we can obtain are 19 weeks gestation week. In these midfetal tissues, neurons in the cortex are still quite immature. In our study, we were able to culture them in healthy form for 10 days. However brain slices start to deteriorate after 20 days of culture. We agree that culturing longer time will help us to answer the developmental question raised by the reviewer. Since human brain development takes 280 days, significantly longer culture time will be needed to observe the maturation changes, which we are not able to do due to limitation of in vitro maintenance of these brain slices.

On the other hand, in our published paper (*Shen et al, Nat Neurosci. 2019; PMID: 30742117*), we transplanted neural progenitor cells (NPCs) differentiated from control and FXS-patient-derived iPSCs into the cortex of immune-deficient neonate mouse brains and analyzed at **4 months** post-transplantation. We found that FXS human neurons developed in transplanted mouse brains exhibited impaired dendritic maturation.

d) The sample image we chosen may not be a good representative of the average of the shFMR1 group. To confirm our results, we went through all of our neuronal tracing data and identified images that can better represent our results. Please see new Fig. 8g. As reviewer requested, we have provided the following wide-field images (scale bar: 100um) for the reviewer (not included in the figures). We want to emphasize that we use confocal z-stacks to perform neuronal tracing in 3D through NeuroLucida 3D Module software (MicroBrightField, Inc) as we have published

previously (Guo, *Nat Medicine* 2011; Shen *Nat Neuroscience* 2019, etc) .Therefore, we could clearly distinguish overlapping neurites.

7) From the methods and results is not abundantly clear when the iPSCs or slices were routinely taken for imaging. There is evidence (especially in FXS) that neurite outgrowth is slower than control neurons (Doers et al., 2014) – if you tested say a week later – would these deficits be normalised out and would the ASD models have caught up with control? Including **1 time point later** in neuronal development would crucial to tie in with existing literature where dendritic structure in *Fmr1*-KO rodent models reaches typical lengths early in development (e.g. Bureau et al., 2008; Till et al., 2015).

Response: We have included description of timeline for all experiments in Methods and timeline schematics for most of the experiments. In the revision, we make sure that all the timings of analysis are included in schematics and figure legend.

We have tried to culture the ASD and control iPSC-derived neurons for longer period. However, the neurons formed clusters connected by neurite bundles which makes it impossible to trace dendrites of individual neurons. Please see following sample images: (scale bar: 50um). As stated above, in our previous publication (Shen et al, *Nat Neurosci.* 2019), human FXS neurons developed in mouse brains did not catch up with the control neurons even at 4 months post-transplantation. The analysis performed by Doers et al 2014 were on axon outgrowth at 2-days after in vitro differentiation. They did not analyze dendrites nor longer period of differentiation.

8) As far as I can tell there is **no direct comparison** made between the dendritic morphology of 5q13.2trip neurons derived from iPSCs to control neurons – but rather the effect of individual genes found in this locus. If the logic is that this genetic mutation enhances MAP1B levels, one might expect that the dendritic lengths of these neurons differ to control. By eye, judging to the total neurite (dendritic) lengths between control neurons (**Figure 1G**) and 5q13.2trip neurons (**Figure 2G**) there is no difference in total length. This comparison is important and may impact the authors overall conclusion about the role of MAP1B enriched expression on neuronal structure. If 5q13.2trip neurons express endogenously higher levels of MAP1B it would be expected that their **dendritic structure is more simplified than that of control neurons** - based on the central hypothesis of this study. Simply looking at each of the genes in turn from this locus may overlook potential polygenic effects.

Response: We want to clarify that human PSC lines can have diverse features and phenotypes, resembling the diversity of human population. Therefore one control PSC line may not be the same as another control PSC line. Isogenic controls are therefore more informative for assessing impact of disease mutations and gene gain- and loss-of functions. In addition, culture conditions and assay conditions can also vary over time, to obtain rigorous results, experimental and control neurons must be differentiated, treated, and analyzed at the same time to be compared.

The figure 1G is to compare idCAS9-A hESC-differentiated neurons with MAP1B-EE (LV-sgMAP1B) to isogenic control idCAS9-A neurons without MAP1B-EE (LV-sgNC). Therefore the “healthy” control neurons in this figure were idCAS9-A hESC differentiated neurons infected with LV-sgNC.

The figure 2G is to compare ASD neurons differentiated from 5q13.2Trp ASD patients that do not have *MAP1B* gene knockdown (LV-shNC) to the ASD neurons that have *MAP1B* gene

knockdown (LV-shMAP1B). The “healthy” control neurons in this figure are neurons differentiated from 5q13.2Trp ASD iPSCs infected with LV-shMAP1B.

As you can see that in this two figures, the “healthy” control neurons have similar dendritic complexity and length.

To further address reviewer’s concerns, we have directly compared between the dendritic morphology of 5q13.2trip neurons to neurons differentiated from health control iPSCs (with LV-GFP infection to allow for neuronal tracing), please see fig. S16 a-c. The results confirmed 5q13.2trip neurons indeed have reduced dendritic complexity and length when compared to control neurons.

Minor:

1) Dendrite analysis: Did the authors confirm that the structures they reconstructed were actually dendrites – or are these neurites comprising both axonal and dendritic sections. This distinction should be stated.

Response: We only traced dendrites in this study. Axons have significantly different morphology and marker expression from dendrites and we have confirmed our criteria using MAP2 (dendritic marker) staining. Axons are significantly longer and thinner. In addition, the thickness of axon is uniform throughout the length, while dendrites are thicker to thin along from their starting to ending points. Furthermore, axons branch at the distal ending only, whereas dendrites branch along the entire length.

2) Mini EPSC analysis: the phrase “Threshold was typically kept at -5 pA and adjusted with respect to the baseline noise.” Is quite nebulous. What was the average threshold used for each group and did it differ

between tested groups? A minor arbitrary shift in threshold can wildly shift the number of events recorded, which may bias the data.

Response: we have provided more information about the analysis and modified the statement in the Methods section: Events of magnitude less than three times of the standard deviation ($\text{std} \times 3$) of baseline or less than 5 pA were rejected. We provide the figures below for the reviewer. We do not feel these quality control data need to be included in the manuscript but we can if editor recommends.

3) There has been much literature on the excitability of neurons in Fmr1 mice and rats during early postnatal development (e.g. Gibson et al., 2009, Pilpel et al. 2009, Luque et al., 2017 Domanski et al., 2019, Booker et al., 2019, Ordemann et al., 2021...), it might be useful to highlight some of these studies and causally link them to the physiological correlations you have made between synaptic function and MAP1B expression.

Response: We thank the reviewer to bring up these literatures, which are very useful. We highlighted most of the studies in our paper.

4) The introduction is very FXS focused and the 5q13.2trip approach is very minimally discussed. A more general background would be useful, such as the detail at the start of the second results section.

Response: We have added more information about the 5q13.2trip in the Result section.

Reviewers' Comments:

Reviewer #1:

Remarks to the Author:

The authors have fully addressed my concerns. They are now providing convincing data on the activity of the autolysosomal system.

Reviewer #2:

Remarks to the Author:

The authors have addressed my primary concerns